# Vapor plumes in a tropical wet forest: spotting the invisible evaporation

César Dionisio Jiménez–Rodríguez[1,2], Miriam Coenders–Gerrits[1], Bart Schilperoort[1], Adriana del Pilar González–Angarita[3], and Hubert Savenije[1]

[1]Delft University of Technology. Water Resources Section. Stevinweg 1, 2628 CN Delft, The Netherlands.
[2]Tecnológico de Costa Rica. Escuela de Ingeniería Forestal. 159-7050, Cartago, Costa Rica.
[3]Independent Researcher

**Correspondence:** César Dionisio Jiménez–Rodríguez (cdjimenezcr@gmail.com)

**Abstract.** Forest evaporation exports a vast amount of water vapor from land ecosystems into the atmosphere. Meanwhile, evaporation during rain events is neglected or considered of minor importance in dense ecosystems. Air convection moves the water vapor upwards leading to the formation of large invisible vapor plumes, while the identification of visible vapor plumes has not been studied yet. This work describes the formation process of vapor plumes in a tropical wet forest as evidence of evaporation processes happening during rain events. In the dry season of 2018 at La Selva Biological Station (LSBS) in Costa Rica it was possible to spot visible vapor plumes within the forest canopy. The combination of time–lapse videos at the canopy top with conventional meteorological measurements along the canopy profile allowed us to identify the driver conditions required for this process to happen. This phenomenon happened only during rain events. Visible vapor plumes during day time occurred in the presence of precipitation ($P$), air convection identified by the temperature gradient ($\frac{\Delta \theta_v}{\Delta z}$) at 2 m height, and a lifting condensation level at 43 m height ($z_{\mathrm{lcl.43}}$) smaller than 100 m.

## 1  Introduction

Forest cover in tropical regions is endangered by deforestation (Curtis et al., 2018; Rosa et al., 2016), compromising the evaporation flux from land. Forest evaporation is a mixture of water vapor originated from water intercepted on plant surfaces, soil water and plant transpiration (Roberts, 1999; Savenije, 2004; Shuttleworth, 1993). Forest evaporation is considered of major importance as a regional and local cooling system (Ellison et al., 2017) as a result of their capacity to recycle the atmospheric moisture at different time scales (van der Ent and Savenije, 2011). The water vapor originated from evaporation at the surface is horizontally transported in the atmosphere by advection (Lavers et al., 2015; Strong et al., 2007), where the forest presence at continental scale induced the "biotic pump mechanism" that favored the maintenance of similar precipitation amounts between inland and coastal environments (Makarieva and Gorshkov, 2007; Makarieva et al., 2013a). Meanwhile, the vertical transport is linked to wind shear (Chen et al., 2015) and convection (Trzeciak et al., 2017) that in large ecosystems influence the formation of convective clouds at the top of the atmospheric boundary layer (Fuentes et al., 2016; Manoli et al., 2016). This process plays an important role in the formation of precipitation in tropical basins (Adams et al., 2011; van der Ent and Savenije, 2011),

because of the contribution of water vapor originated from local evaporation (Brubaker et al., 1993).

Evaporation is usually neglected or considered of minor importance during rain events in dense forest ecosystems (Klaassen et al., 1998). This is because during rainfall the vapor pressure deficit is close to zero (Bosveld and Bouten, 2003; Loescher et al., 2005; Mallick et al., 2016), reducing the atmospheric water demand and stopping the transpiration process (Gotsch et al., 2014). However, the increment of evaporation with the size of rain events suggest that evaporation also occurs during the events and not only afterwards (Allen et al., 2020). This has been evidenced by discrepancies found between modelled and measured evaporation rates in tropical forests (Schellekens et al., 2000). When it rains part of the precipitation is intercepted and evaporated directly to the atmosphere (David et al., 2006), even when vapor pressure deficit and available radiation are low (Lankreijer et al., 1999). Under high humidity conditions a portion of the precipitation can evaporate after a raindrop splashes on the canopy or the forest floor. This process is known as "splash droplet evaporation" (Dunin et al., 1988; Dunkerley, 2009; Murakami, 2006) and is based on the principle that raindrop size increases with rain intensity. Consequently, when larger drops hit the surface (e.g, ground, leaves, branches) allow the formation of smaller rain droplets that can be easily evaporated after the splash. This process has been pointed out as the main source of evaporation to explain the difference between intercepted water and measured evaporation in studies carried out in banana plants (Bassette and Bussière, 2008) and Eucalyptus plantations (Dunin et al., 1988).

Forest evaporation produces coherent structures of water vapor called plumes, cells, or rolls (Couvreux et al., 2010). Plumes of water vapor have been identified above forest ecosystems during day time with high resolution scanning Raman LIDAR technique (Cooper et al., 2006; Kao et al., 2000). These plumes reached heights above the canopy up to $100\,\mathrm{m}$, depicting their importance as water vapor providers at local scale. This phenomenon has been studied in astrophysics (Berg et al., 2016; Sparks et al., 2019), vulcanology (Kern et al., 2017; Sioris et al., 2016), regional, and global meteorology (Herman et al., 2017; Knoche and Kunstmann, 2013; Wang, 2003; Wright et al., 2017). However, to the authors best knowledge little attention has been drawn to small events observed during rain events. Additionally, Couvreux et al. (2010) highlighted the lack of sampling techniques being able to characterize the occurrence of these plumes close to the surface. Visible vapor plumes are classified as ascending clouds formed by clusters of tiny particles of water in liquid form (Spellman, 2012). This characteristic makes difficult to measure them with sophisticated systems based on 3D wind components (e.g., eddy–covariance systems) that are developed to measure water in gas form (Foken et al., 2012a). This type of measurements are sensitive to rainy and high humidity conditions (Camuffo, 2019; Foken et al., 2012b; Kelton and Bricout, 1964; Moncrieff et al., 2005; Mauder and Zeeman, 2018; Peters et al., 1998) making it difficult to use them to identify the occurrence of visible vapor plumes in forested ecosystems. This mismatch between measurement systems and target phenomena, underlines the need to identify the conditions under which visible vapor plumes are formed. This type of constraints requires an innovative data analysis approach, which is the focus of this paper. This work aims (1) to test an innovative approach to link visual information and conventional meteorological data describing a local hydrological phenomenon. Also, (2) to identify the meteorological conditions when visible vapor plumes are present in a Tropical Wet Forest, and it tries (3) to explain the processes involve on the formation

of these plumes. The data analysis is based on conventional meteorological data vertically distributed along the forest canopy layer and time-lapse videos during day-time conditions.

## 2  Methodology

### 2.1  Study Site

The monitoring was carried out at La Selva Biological Station (LSBS) on the Caribbean lowlands of Costa Rica (N: $10°26'0''$ – W: $83°59'0''$). This station registered a mean annual precipitation of $4351\,\text{mm}\,\text{yr}^{-1}$, a mean annual temperature of $26.3\,°C$, and a mean daily temperature difference of $9.5\,°C$. A short dry season is present in LSBS between February and April every year, and it is characterized by a reduction in the precipitation without vegetation experiencing a soil water deficit (Sanford Jr. et al., 1994; Lieberman and Lieberman, 1987; Loescher et al., 2005). LSBS is covered by a matrix of old growth and secondary forests, small forest plantations, and experimental permanent plots with mixed tree species (Figure 1). All the instrumentation was placed at the Major Research Infrastructure plot (MRI–plot) of 1.0 ha, located within an old growth forest on the upper terrace of the Puerto Viejo river (Sanford Jr. et al., 1994). The MRI–plot is situated in the upper section of a small hill facing South–West towards an affluent of the Puerto Viejo river. The soil is classified as Andic Humidotropept with a clay and organic matter content of $35\,\%$ and $23\,\%$, respectively (Sollins et al., 1994). Tree density in 2017 was $371\,\text{trees}\,\text{ha}^{-1}$ of individuals with a tree diameter bigger than $10\,\text{cm}$. The palm *Welfia regia* H.Wendl and the tree *Pentaclethra macroloba* (Willd.) Kuntze are the most abundant species with $56\,\text{trees}\,\text{ha}^{-1}$ and $43\,\text{trees}\,\text{ha}^{-1}$, respectively. Average leaf area index (LAI) in 2005 was $3.56\,\text{m}^2\,\text{m}^{-2}$ (Tang et al., 2012). The plot is located within a stable forest plot in terms of changes in canopy height and tree biomass fixation (Dubayah et al., 2010).

### 2.2  Experimental Design

The monitoring was carried out on the MRI–plot in the highest tower (43 m), which is located within a depression of the forest canopy (Figure 1). Along the vertical axis of the tower, the air temperature ($°C$) and relative humidity ($\%$) were measured with HOBO® smart sensors (part code: S-THB-M008). The sensors were located at $2\,\text{m}$, $8\,\text{m}$ and $43\,\text{m}$ height, placed at a distance of $1.5\,\text{m}$ from the tower and protected with a radiation shield (HOBO® part code: RS-3) of $10\,\text{cm}$ diameter. The use of radiation shields together with conventional air temperature sensors allows keeping a mean absolute error during day time in warm tropical ecosystems below $0.3\,°C$ (da Cunha, 2015; Terando et al., 2017). Also, the shelter provided by the forest canopy for the measurements carried out at $2\,\text{m}$ helps to record similar temperatures to the surrounding near-surface environment (Lundquist and Huggett, 2008). The measurement of minimum air temperatures or night-time temperatures does not require the cover of the radiation shield to keep low biases ($<0.5\,°C$) on the mean air temperature due to the reduced or total absence of solar radiation (Terando et al., 2017). At the highest point of the tower, the precipitation ($\text{mm}\,\text{min}^{-1}$) was

recorded with a Davis® rain gauge. Soil temperature (°C) was measured in two different locations at 5 cm and 15 cm depth with a soil temperature sensor (HOBO® part code: TMC20-HD). Soil moisture ($\Theta$, $m^3\,m^{-3}$) was measured at the same locations as soil temperature at 5 cm depth with an $ECH_2O$® EC sensor. Soil temperature was recorded with a 4-channel data logger (HOBO® part code: U12-008) and the other sensors with a USB Micro Station (HOBO® part code: H21-USB). Meteorological data collected along the tower and soil temperature data were recorded with 1 min and 5 min average, respectively. All data was summarized in 5 min time intervals for the analysis. A Bushnell® Natureview® Essential HD camera (12 megapixels) was installed at the top of the tower facing North–West.

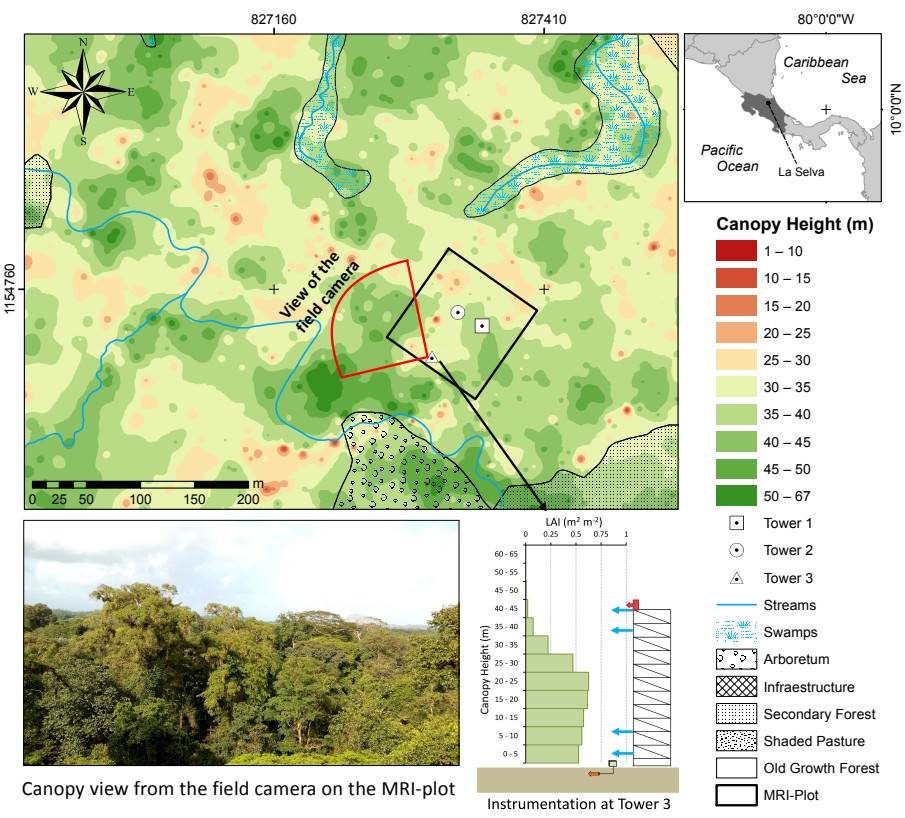

**Figure 1.** Canopy height and land cover map of the area surrounding the Major Research Infrastructure plot (MRI–plot) at La Selva Biological Station, Costa Rica. The photograph in the map shows a view from the field camera at the top of the tower.

## 2.3   Monitoring Period

All environmental variables were monitored between 2018-01-24 and 2018-03-26. The camera was installed to collect photographs above the canopy between 2018-03-21 and 2018-03-25. The photographs were set to be collected continuously from 5:00 to 18:30 hours local time (UTC-6). However, the light conditions affected the images selected as suitable for analysis

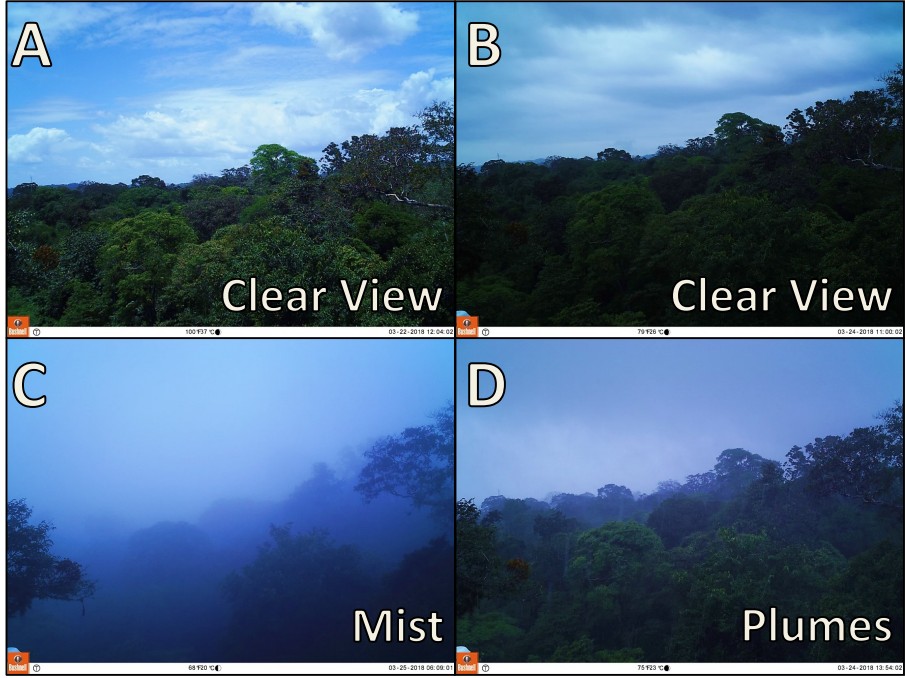

**Figure 2.** Visual monitoring showing the 3 conditions used to classify the canopy photographs on the time-lapse videos. The pictures A and B show the "Clear View" classification, picture A on a sunny day and picture B during rain. Picture C describes the Mist and picture D shows the plumes rising from the forest canopy.

(see Appendix B). These pictures were used to determine the timing when the vapor plumes were visible at the MRI–plot. The photographs were classified into three conditions (Figure 2):

- Clear View: includes all the pictures with clear and cloudy sky where the canopy is clearly visible and there is neither mist nor plumes present (Figure 2 A and B).

- Mist and Fog: includes the presence of a homogeneous blurry view of the canopy. The blurriness of each picture varies depending on the humidity conditions. Special care was taken to prevent the erroneous classification of photographs affected by a fogged–up lens. This category is called "mist" from now onwards (Figure 2 C).

- Plumes: includes the presence of buoyant vapor clouds risen from the forest canopy (Figure 2 D). These cloud bodies change their vertical position in consecutive frames. Rising vapor plumes can be observed in the online video of 2018-03-24 available at https://doi.org/10.4121/uuid:997cc9d8-2281-453e-b631-5f93cfebe00e (Jiménez-Rodríguez et al., 2019b).

## 2.4 Data Analysis

Data processing and analysis was performed with the open source software R (R Core Team, 2017). All temperatures were converted from K to °C. Superficial soil temperature ($T_{s.0}$, °C) was estimated with equation 1 (Holmes et al., 2008). This equation describes the diurnal variations of soil temperature as sine waves depending on the 24 h moving averages of soil temperature at 5 cm depth ($T_{s.5}$, °C). The daily amplitude of air temperature ($T_A$, °C) is defined as the difference between $T_{s.5}$ and the air temperature at 2 m ($T_{2m}$). The oscillations are determined by the damping depth ($\nu$, m) which is calculated with equation 2. Depth difference between the $T_{s.0}$ and $T_{s.5}$ is defined as $z_b$ (m). The sine pattern depends on the angular frequency ($\omega$, s$^{-1}$), time ($t$) in s and $\phi$ (-) as a constant for phase change. Equation 3 is used to determine $\omega$ with $\tau$ (s) as the wave period. Equation 2 calculates $\nu$ with the soil thermal diffusivity ($\eta$, m$^2$ s$^{-1}$) and $\omega$. Equation 4 (Nakshabandi and Kohnke, 1965) is used to determine $\eta$, where $\rho_s$ is the soil bulk density of 0.76 g cm$^{-3}$ (Sollins et al., 1994) for the experimental plot, $c_s$ is the specific heat for clay soils (837.36 W kg$^{-1}$ °C$^{-1}$) and $k$ is the soil thermal conductivity of 1.58 W m$^{-1}$ °C$^{-1}$ (Pielke, 2013). These last two parameters were chosen according to the soil water conditions during the monitoring period, which was close to soil field capacity (see Appendix B).

$$T_{s.0} = T_{s.5} + T_A \, e^{(\frac{-z_b}{\nu})} \sin(\omega t - \frac{z_b}{\nu} + \phi) \tag{1}$$

$$\nu = \sqrt{\frac{2\eta}{\omega}} \tag{2}$$

$$\omega = \frac{2\pi}{\tau} \tag{3}$$

$$\eta = \frac{k}{\rho_s c_s} \tag{4}$$

Virtual potential temperature ($\theta_v$, °C) of the air was calculated to take into account the variation in the adiabatic lapse rate due to changes in pressure (Barr et al., 1994; Stull, 1988, 2017). For saturated (cloudy) air conditions equation 5 calculates the $\theta_v$ based on the water-vapor mixing ratio ($\psi_s$) of the saturated air, the liquid water mixing ratio ($\psi_L$) and the virtual temperature ($\theta$). The parameters $\psi_s$ and $\psi_L$ were determined with equations 7 and 8, respectively. These equations requires to know the mass of the liquid water in the air ($m_{liq.air}$), the mass of the water vapor in the air ($m_{vap.air}$), and the mass of the dry air ($m_{dry.air}$). Due to the lack of instrumentation to estimate the mass of liquid water in the air, we used a fixed value of 0.05 g m$^{-3}$. This value corresponds to the liquid water content (LWC) in the air reported by Thompson (2007) for continental fog events. The selection of this value was based on (1) the similarity between the vapor plumes and fog, and (2) because both types of events occur close to the ground surface. The variables $m_{vap.air}$ and $m_{dry.air}$ were determined using the saturation and actual vapor

pressures of the air (Stull, 2017). The virtual temperature was estimated with equation 6 where $\Gamma_{\mathrm{d}}$ is the dry adiabatic lapse rate near the surface ($0.0098\,^{\circ}\mathrm{C\,m^{-1}}$), $z$ is the height above the ground in m and $T_{\mathrm{z}}$ is the air temperature at the same heights.

$$\theta_{\mathrm{v.z}} = \theta_{\mathrm{z}}(1 + 0.608\,\psi_{\mathrm{s}} - \psi_{\mathrm{L}}) \tag{5}$$

$$\theta_{\mathrm{z}} = T_{\mathrm{z}} + \Gamma_{\mathrm{d}}\,z \tag{6}$$

5 $$\psi_{\mathrm{s}} = \frac{m_{\mathrm{vap.water}}}{m_{\mathrm{dry.air}}} \tag{7}$$

$$\psi_{\mathrm{L}} = \frac{m_{\mathrm{liq.water}}}{m_{\mathrm{dry.air}}} \tag{8}$$

Convection can be identified by evaluating the temperature gradient ($\frac{\Delta\theta_{\mathrm{v}}}{\Delta z}$) due to the absence of wind profile measurements to determine the atmospheric stability parameter along the tower. Values of $\frac{\Delta\theta_{\mathrm{v}}}{\Delta z} > 0$ are linked to stable stratification, meanwhile $\frac{\Delta\theta_{\mathrm{v}}}{\Delta z} < 0$ show an unstable stratification (Stull, 2017), which will drive convection.

The condensation of vapor close to the forest canopy can be identified by calculating the lifting condensation level ($z_{\mathrm{lcl}}$) in m with equation 9. This equation determines the elevation at which a parcel of air condensates allowing the formation of clouds. This equation uses the difference between air temperature ($T_{\mathrm{z}}$) and dew point temperature ($T_{\mathrm{dew.z}}$) at one specific height ($z$), divided by the difference between $\Gamma_{\mathrm{d}}$ and the dew point temperature lapse rate ($\Gamma_{\mathrm{dew}}$) (Stull, 2017).

15 $$z_{\mathrm{lcl}} = \frac{T_{\mathrm{z}} - T_{\mathrm{dew.z}}}{\Gamma_{\mathrm{d}} - \Gamma_{\mathrm{dew}}} \tag{9}$$

An estimation of the evaporation during the monitored period was retrieved from Jiménez-Rodríguez et al. (2020). This data set is used only as a reference of the evaporation process during the monitoring period on the same site. This is because this quantification has limitations accomplishing the Monin-Obukhov similarity (MOST) theory for complex terrains (Breedt et al., 2018). So, it is based only on the vertical transport of water vapor, neglecting the advected energy of the forest canopy.

## 3   Results and Discussion

The monitoring period experienced a diurnal variation in air temperature along the vertical profile of the canopy, with a temperature difference of more than $10\,^{\circ}\mathrm{C}$ at $43\,\mathrm{m}$ and less than $7\,^{\circ}\mathrm{C}$ at $2\,\mathrm{m}$ height (Figure 3). The highest temperatures were registered at $43\,\mathrm{m}$ height reaching more than $30\,^{\circ}\mathrm{C}$, decreasing in magnitude towards the forest floor. These peak temperatures

were recorded around noon with differences up to $5\,^{\circ}\mathrm{C}$ between the air temperature at 43 m and 2 m height. The $T_{\mathrm{s.0}}$ oscillates between $20.7\,^{\circ}\mathrm{C}$ and $25.4\,^{\circ}\mathrm{C}$. The amplitude of the oscillation increased with the sunniest days but the daily difference does not exceed the $4\,^{\circ}\mathrm{C}$. The maximum $\Theta$ value was $0.47\,\mathrm{m^3\,m^{-3}}$ during the heavy rains, almost reaching the saturation point for clay soils of $0.50\,\mathrm{m^3\,m^{-3}}$ (Saxton and Rawls, 2006). The minimum $\Theta$ was recorded after the driest period just before

5 the rains on 2018-03-24 ($0.42\,\mathrm{m^3\,m^{-3}}$) getting close to soil field capacity for clay soils (Saxton and Rawls, 2006). Evaporation always occurs during daytime on all sampling days (Figure 3). During the four sunny days the evaporation was larger than $5\,\mathrm{mm\,d^{-1}}$, with a contribution of more than $1.0\,\mathrm{mm\,d^{-1}}$ from 8 m height and no more than $0.7\,\mathrm{mm\,d^{-1}}$ from 2 m height (Jiménez-Rodríguez et al., 2020). In contrast, during 2018-03-24 the continuous rains sum up $58.7\,\mathrm{mm\,d^{-1}}$ and the evaporation was estimated as $1.8\,\mathrm{mm\,d^{-1}}$ at 43 m and only $0.2\,\mathrm{mm\,d^{-1}}$ at 2 m height (Table 1).

**Table 1.** Daily summary of precipitation and evaporation at 43 m, 8 m, and 2 m height according to Jiménez-Rodríguez et al. (2020) for the experimental site during the monitoring period.

| Date | Precipitation ($\mathrm{mm\,d^{-1}}$) | Evaporation ($\mathrm{mm\,d^{-1}}$) | | |
|---|---|---|---|---|
| | | 0–43 m | 0–8 m | 0–2 m |
| 2018-03-21 | 0.0 | 6.0 | 1.5 | 0.7 |
| 2018-03-22 | 0.0 | 5.4 | 1.1 | 0.4 |
| 2018-03-23 | 4.6 | 5.8 | 1.1 | 0.3 |
| 2018-03-24 | 58.7 | 1.8 | 0.5 | 0.2 |
| 2018-03-25 | 0.0 | 5.3 | 1.2 | 0.5 |

Note: all evaporation values corresponds to the water vapor produced from
the forest floor up to the specified height.

During the visual monitoring with the field camera, clear view conditions were predominant along four days (Figure 3). These days were characterized by sunny conditions with temperatures above the $25\,^{\circ}\mathrm{C}$, no large rain events and a decreasing trend in soil moisture. These days were characterized by cumulus clouds crossing the sky above the forest canopy in day time. Any water vapor ascending from the forest canopy will need to reach a height of more than 100 m to form visible vapor plumes

15 (Figure 3). Also, on 2018-03-24 it was possible to identify three short periods with clear view conditions in between the rains. Mist formation was identified on 2018-03-23 and 2018-03-25 before 7:00 a.m. Mist might be formed early in the morning during the sampling dates 2018-03-21 and 2018-03-22. However, the time lapse video did not work at those times (Table B1). These mist events were linked with superficial soil temperatures higher than $2\,^{\circ}\mathrm{C}$ with respect to air temperature. Finally, the vapor plumes were visible only during rainy conditions on 2018-03-24 (videos available at Jiménez-Rodríguez et al. (2019b)).

20 Soil temperatures during this day were warmer than the air column along the forest canopy (Figure 3).

Evaporation during sunny days provided the conditions to form vapor plumes as those ones described by Cooper et al. (2006) and Kao et al. (2000). The evaporation peaks during these days occurred around noon, registering a $z_{lcl}$ higher than 500 m (Figure 3) which is the height required to form clouds and be visible. This is the reason why is not possible to see the vapor rising from the surface. The vapor plumes were visible on the day with continuous precipitation (2018-03-24). On this day, the $z_{lcl}$ dropped beneath 100 m because during rain events the $\theta_v$ of all the air column dropped quickly. This drop kept the $\theta_v$ beneath the superficial soil temperature, allowing a localized convection event. This convection process forced the evaporated water to move upwards forming buoyant clouds close to the forest surface. The evaporation during rain events is the result of the splash droplet evaporation process (Murakami, 2006; Dunkerley, 2009), which can provide water vapor as a consequence of the fragmentation of raindrops when hitting the surface.

Energy convection plays an important role in forest ecosystems during night time (Bosveld et al., 1999). This is a consequence of the mass transport capacity of the intermittent nocturnal convective fluxes (Cooper et al., 2006). The convection process is forced by the ground heat flux (Jacobs et al., 1994), which is enhanced by the larger soil moisture in the clay soil which increases the soil heat capacity (Abu-Hamdeh, 2003). A coupled canopy system enables sensible heat and water vapor transport from the soil to the atmosphere just above the canopy layer (Göckede et al., 2007). This facilitates the generation of the convection process, allowing the ascending warm air to cool down at the canopy top and condensate forming the visible water vapor plumes. The condensation releases heat (Goosse, 2015), driving the convection. Vapor plumes are always present as a consequence of the moisture exchange between the surface and the atmosphere (Lawford, 1996), where evaporation from land covers with enough water supply provides the required air moisture (Kao et al., 2000). However, the conditions needed to form a visible buoyant cloud close to the surface require a big difference in air temperature over height. Temperature gradient at 43 m, 8 m and 2 m is negative during plumes and mist conditions, meanwhile clear view conditions has a larger range with more positive values (see Appendix C).

The visible vapor plumes can be spotted on the canopy depressions surrounding the tower (Figure 1). These depressions are characterized by a low leaf area index and shorter canopy height, which translates into areas with low potential to produce transpiration during rain events. This implies that the main source of water vapor is linked to water evaporated from wet surfaces and soil evaporation, while transpiration may contributes to a lesser extent. Visible vapor plumes are the result of the condensation of water vapor rising from a warmer surface. When a column of warm humid air reaches the dew point temperature, the water vapor condensates around aerosols in the air allowing the formation of clouds (Stull, 2017). In this regard, there are different sources of aerosols at LSBS. One source is linked to wind carrying aerosols from nearby agricultural land uses (Loescher et al., 2004). A second source is linked to convective rains that characterize the dry season at LSBS. Additionally, it cannot be discarded the presence of bioparticles (e.g., airborne bacteria, fungi, pollen, plant fragments) as a source of aerosols from the forests (Huffman et al., 2013; Valsan et al., 2015). The high intensity rains may induce the bioparticles burst from the forest canopy. This bioparticles have been reported in Australia (Bigg et al., 2015), India (Valsan et al., 2015), Mexico (Rodriguez-Gomez et al., 2020), and the Amazon (Pöschl et al., 2010). Also, convective rains transport from the free

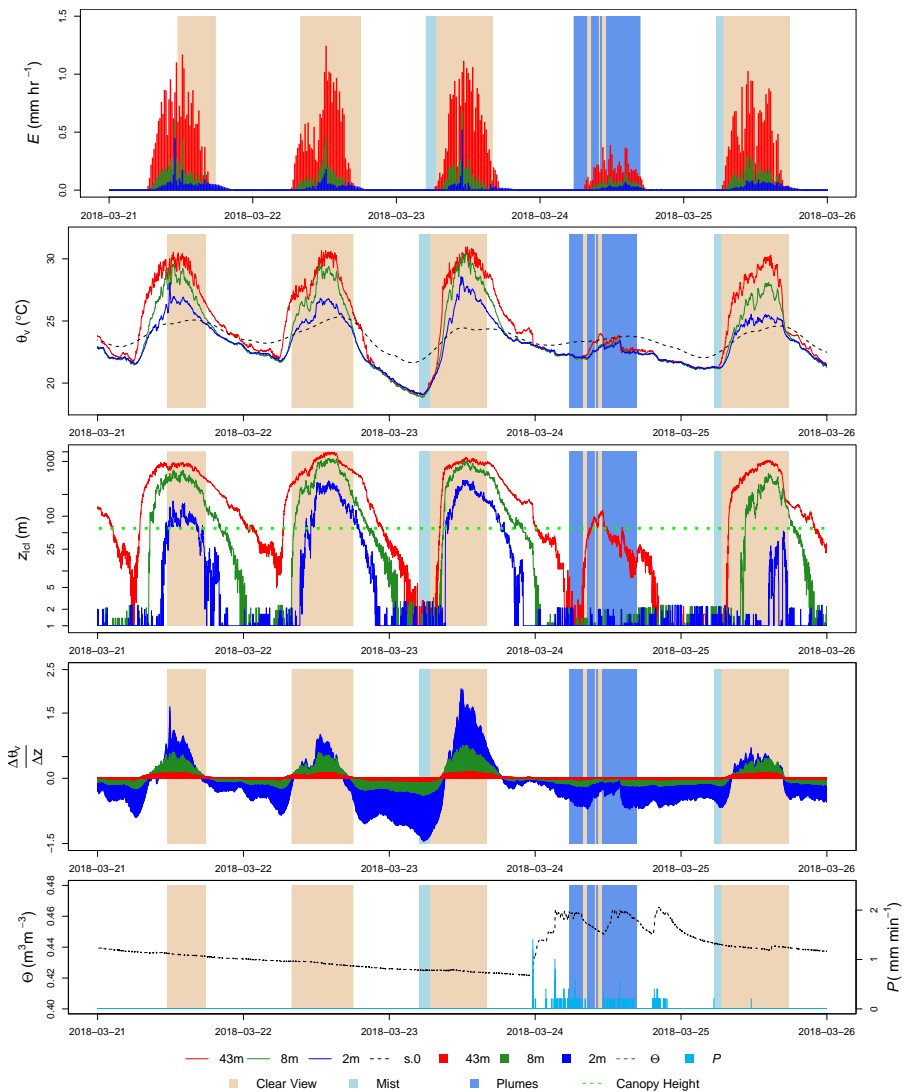

**Figure 3.** Virtual potential temperature ($\theta_v$), lifting condensation level ($z_{lcl}$) in an untransformed semi–logarithmic scale and temperature gradient ($\frac{\Delta\theta_v}{\Delta z}$) at 43 m, 8 m and 2 m height. Additionally, precipitation ($P$) and soil moisture ($\Theta$) are also shown during the visual monitoring between 2018-03-21 and 2018-03-25. Evaporation ($E$) measurements were retrieved from Jiménez-Rodríguez et al. (2020). Background colored areas denoted the three categories in which the photographs were classified: Clear View, Mist and Plumes.

troposphere into the boundary layer a portion of the required aerosols for the condensation process and later they form clouds (Wang et al., 2016). Meanwhile the "splash droplet evaporation" process (Murakami, 2006) provides the main source of water vapor after rain drops hit the canopy and soil surfaces. As plumes are not stagnant and continue moving upwards thanks to air convection, the water vapor is removed from the understory towards higher altitudes. The water condensation at the canopy

level reduced drastically the volume of water vapor due to the phase change (Makarieva et al., 2013b). This allowed the ambient air to remain unsaturated and keeping the "splash droplet evaporation" process providing continuously more water vapor.

Cloud formation usually happens high above the surface boundary layer where the forest canopy is located, but available information of cloud formation close to the forest canopy is scarce. The temperature gradient ($\frac{\Delta\theta_v}{\Delta z}$) at 43 m, 8 m and 2 m is

negative during plumes and mist conditions, meanwhile clear view conditions have a larger range with more positive gradients. Lifting condensation level is a key element that allowed to differentiate between plumes and mist conditions (see Appendix C). The combination of variables such as $z_{lcl}$, $\frac{\Delta\theta_v}{\Delta z}$, and $P$ allows to identify the formation of vapor plumes in Tropical Wet Forests (Figure 4). The $z_{lcl}$ is the height in the atmosphere at which a parcel of moist air becomes saturated if experience a forced ascent (Stull, 2017). It provides an estimate of the height at which the clouds can be formed. The temperature gradient is an

indicator of how easily a parcel of air can be lifted (Spellman, 2012) and can be used as a proxy of the atmospheric stability. During unstable atmospheric conditions ($\frac{\Delta\theta_v}{\Delta z} < 0$) it is easier for the air parcels to move upwards than under stable conditions ($\frac{\Delta\theta_v}{\Delta z} > 0$). Finally, precipitation saturates the air column and provides the water vapor after the splash droplet evaporation process on the canopy and forest floor surfaces.

During the full monitoring period at La Selva Biological Station, only 1.4 % of our study period accomplished the conditions required for the formation of visible vapor plumes (precipitation, $z_{lcl} < 100$ m and $0 > \frac{\Delta\theta_v}{\Delta z} > -1$). These conditions differ from those needed to form mist. In a tropical wet forest in Costa Rica, fog and mist formation happens before sunrise (Allen et al., 1972). However, fog does not involve the upward convective flux needed for vapor plumes, while mist is affected by this upward convective flux but without rain (Stull, 2017). Vapor plumes are buoyant cloud formations with an identifiable

shape (Spellman, 2012), main characteristics that allow the differentiation from fog and mist events. While mist and fog are formed by microscopic water droplets floating in the air which can reduce the visibility to less than one kilometer in the case of fog or a lesser extent with the mist (Spellman, 2012).

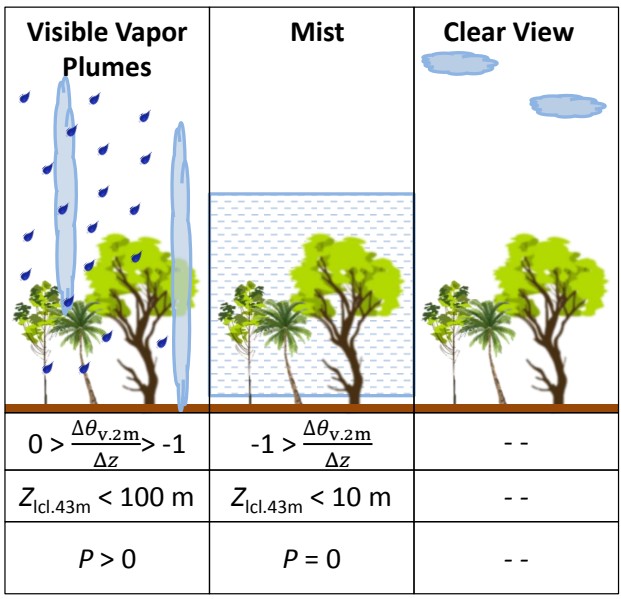

| Visible Vapor Plumes | Mist | Clear View |
|---|---|---|
| $0 > \frac{\Delta\theta_{v.2m}}{\Delta z} > -1$ | $-1 > \frac{\Delta\theta_{v.2m}}{\Delta z}$ | - - |
| $Z_{lcl.43m} < 100$ m | $Z_{lcl.43m} < 10$ m | - - |
| $P > 0$ | $P = 0$ | - - |

**Figure 4.** Simplification diagram describing the required conditions to form visible vapor plumes in a Tropical Wet Forest and the differences between mist and clear view conditions.

This paper described the formation of visible vapor plumes based on photographs as a visual indication of a process that is usually invisible to the human eye. The occurrence of this phenomenon under rainy conditions makes difficult to quantify its contribution to the forest evaporation with current measuring techniques. Vapor plumes occurrence during rainy days compromise the performance of more sophisticated instruments that are highly sensitive to rain or mist conditions (Centre
for Atmospheric Science, 2020; Mauder and Zeeman, 2018). Instruments such as sonic anemometers (e.g., CSAT3, CSAT3B) and Open Path CO2/H2O Analyzers (e.g., LI-7500) are strongly affected by high humidity and rainfall (Campbell Scientific Inc., 2017, 2019; Foken et al., 2012a; LI-COR, 2016; Moncrieff et al., 2005). The presence of rain causes departures from the measurements increasing the sonic speed (Camuffo, 2019; Kelton and Bricout, 1964; Peters et al., 1998) or blocking the face of the transducers (Campbell Scientific Inc., 2017) causing a frequency loss during rain events (Zhang et al., 2016). The
eddy–covariance technique is considered as the standard measurement for determining atmospheric fluxes, however, it is dependent on fully turbulent transport over a homogeneous surface (Foken et al., 2012a). This means that the localized nature of the visible vapor plumes makes measuring them very susceptible to sensor placement, complicating its monitoring using eddy–covariance systems located high above the canopy. Additionally, measuring devices based on 3D wind components (e.g., eddy–covariance systems) are developed to measure water in gas form (Foken et al., 2012a) and are not intended to measure
visible vapor plumes that are ascending clusters of tiny water particles (Spellman, 2012).

The description of the formation process of visible vapor plumes provides a first step on the understanding of this phenomenon within forest hydrology. This description helps to identify the timing when this phenomenon occurs, allowing to

screen existing data sets in other tropical research sites to analyze its frequency of occurrence. However, it is important to test if the conditions required to form visible vapor plumes are the same in other latitudes and ecosystems. Also, new developments in air temperature monitoring techniques such as distributed temperature sensing (Euser et al., 2014; Heusinkveld et al., 2020; Izett et al., 2019; Schilperoort et al., 2018) or thermal infrared imagery (Costa et al., 2019; Egea et al., 2017; Lapidot et al., 2019; Nieto et al., 2019), may contribute to accurately quantify the contribution of visible vapor plumes as local recyclers of forest evaporation. These methods are suitable alternatives to eddy–covariance systems that are sensitive to rainy conditions when visible vapor plumes occur.

Understanding the formation process is a pre-requisite before the quantification of such a complex process. Further studies aiming to analyze in more detail the occurrence of visible vapor plumes will need to consider the conditions that give origin to this phenomenon: air convection, precipitation presence, and lifting condensation level at the top of the canopy. While the quantification of its contribution to the hydrological cycle have to overcome the limitations of current measuring techniques. The identification of air convection should be based on direct measurements of ground heat flux (e.g, soil heat flux plates), a more detailed air temperature profile along the forest canopy (e.g, using distributed temperature sensing), and multiple wind speed measurements along the canopy profile (e.g, at least one per canopy layer plus one above the canopy). This set of measurements will help to identify air convection and advection within the canopy structure. The wind measurements should be carried pairing sonic and cup anemometers at the same locations, allowing to overcome the limitations of sonic anemometers during rain events (Mauder and Zeeman, 2018), when liquid water covers the ultrasonic transducers.

The evaporation contribution to the local hydrological cycle by visible vapor plumes requires a detailed quantification of the latent heat flux ($\rho \lambda E$) above and below the canopy. The use of net radiometers at different heights (same locations as wind speed measurements) will complement the detailed air temperature, and wind speed measurements. It is important to underline that some experimental sites worldwide accomplish the equipment requirements above mentioned (FLUXNET, 2020), opening the opportunity to reanalyze their data sets towards the identification of the conditions needed for the formation of visible vapor plumes. Also, these sites provide an opportunity to quantify the bias that eddy–covariance systems make due to the existence of this phenomenon. Direct measurements of atmospheric water (gas and liquid phase) can be achieved with closed-path gas analyzers (e.g, LI-7000DS-LI-COR, EC155-Campbell Sci., FMA-Los Gatos Research), allowing to determine the total water content in the air. These measurements will benefit from combining high resolution infrared images from above and below canopy, allowing to study the spatial distribution of the phenomenon. These images will provide information under day and night conditions, helping to identify the splash-droplet evaporation process at canopy and ground level when the view field is focused towards specific locations of the forest canopy. Finally, further research can search for the detailed source of vapor with the implementation of direct measurements of water stable isotopes using mass spectrometers or cavity output spectroscopy. This type of research can provide more insights into the effect of vapor plumes on the micro-climate of forest ecosystems. Moreover, the occurrence of this phenomenon in other vegetation types may be addressed to understand the main drivers and

the role played in local hydrological systems.

## 4 Conclusions

The visual monitoring captured the formation of visible vapor plumes close to the surface boundary layer of a Tropical Wet Forest (TWF) during rainy conditions. These visible plumes are the visual evidence of evaporation processes happening during rain events, where the splash droplet evaporation process provides the required water vapor to form visible vapor plumes. This water vapor is part of the intercepted water evaporated from the forest floor and plant surfaces since transpiration is likely reduced by the low vapor pressure deficit but not stopped. It is raised up by air convection driven by warm soil temperatures. Condensing finally close to the forest canopy due to the drop in the virtual potential air temperature along the forest air column. Consequently, this phenomenon can be identified in TWF when precipitation occurs, the lifting condensation level at 43 m height ($z_{\mathrm{lcl}}$) is lower than 100 m, and the temperature gradient ($\frac{\Delta \theta_v}{\Delta z}$) at 2 m height is between 0 and -1 °C m$^{-1}$. Contrary to the vapor plumes, mist appear when no precipitation occurs ($P = 0$), $z_{\mathrm{lcl}}$ at 43 m is less than 10 m and $\frac{\Delta \theta_v}{\Delta z}$ is less than -1 °C m$^{-1}$. This work also brings the attention to the forest evaporation role during rain events, where little information is still available. The exploratory nature of this work, opened new research opportunities aiming to improve the setup to monitor this phenomenon and provide a further accurate quantification of the contribution within the local hydrology.

*Author contributions.* The project conceptualization and funding acquisition was carried out by César Dionisio Jiménez-Rodríguez (CJR) and Miriam Coenders-Gerrits (MCG). The investigation and data curation was carried out by Adriana Gonzalez-Angarita and CJR. Data analysis was performed by Bart Schilperoort and CJR. Finally, the project administration, writing of the original manuscript and data visualization was carried out by CJR with inputs from all the co-authors.

*Competing interests.* The authors declare that they have no conflict of interest.

*Acknowledgements.* This work was carried out with a fellowship from the Organization for Tropical Studies (Glaxo Centroamerica Fellowship–Fund 502). With the aid of a scholarship from PINN-MICITT Costa Rica (contract: PED-032-2015-1) and the aid of the grant 863.15.022 from The Netherlands Organization for Scientific Research (NWO). Also, NASA's funding NNX12AN43H and 80NSSC18K0708 for providing the Leaf Area Index data sets. Special thanks to Bernal Matarrita, Orlando Vargas, Wagner López, Danilo Brenes, Diego Dierick, Enrique Castro and Marisol Luna for their help and advice in the research station. Finally, to all the staff of the OTS for its willingness to support our project and to Shigeki Murakami and one anonymous reviewer, who helped to improve the paper.

*Data availability.* Time lapse videos are available online in the 4TU data repository at https://data.4tu.nl/repository/uuid:997cc9d8-2281-453e-b631-5f93cfebe00e (Jiménez-Rodríguez et al., 2019b). Meteorological data used in this manuscript are available online in the 4TU data repository (https://doi.org/10.4121/uuid:e70993d2-5852-4f63-9aff-39451fbd3fde; Jiménez-Rodríguez et al. (2019a).

## Appendix B: Time-lapse videos detailed information

**Table B1.** Time windows with suitable images for analysis during the 5 sampling days surveyed with the camera.

| Sampling Date | Time Interval | Initial Time | Final Time |
|---|---|---|---|
| 2018-03-21 | 5 minutes | 11:27 | 17:45 |
| 2018-03-22 | 5 minutes | 8:00 | 11:00 |
| 2018-03-22 | 1 minute | 11:00 | 18:00 |
| 2018-03-23 | 1 minute | 5:10 | 16:42 |
| 2018-03-24 | 1 minute | 5:30 | 16:42 |
| 2018-03-25 | 1 minute | 5:32 | 17:38 |

Note: the change of sampling intervals from 5 minutes to 1 minute was carried out the second day of video monitoring aiming to improve the quality of the survey. The camera was set to take images from 5:00 to 18:30, the time windows showed in the table correspond to the period with images suitable for analysis.

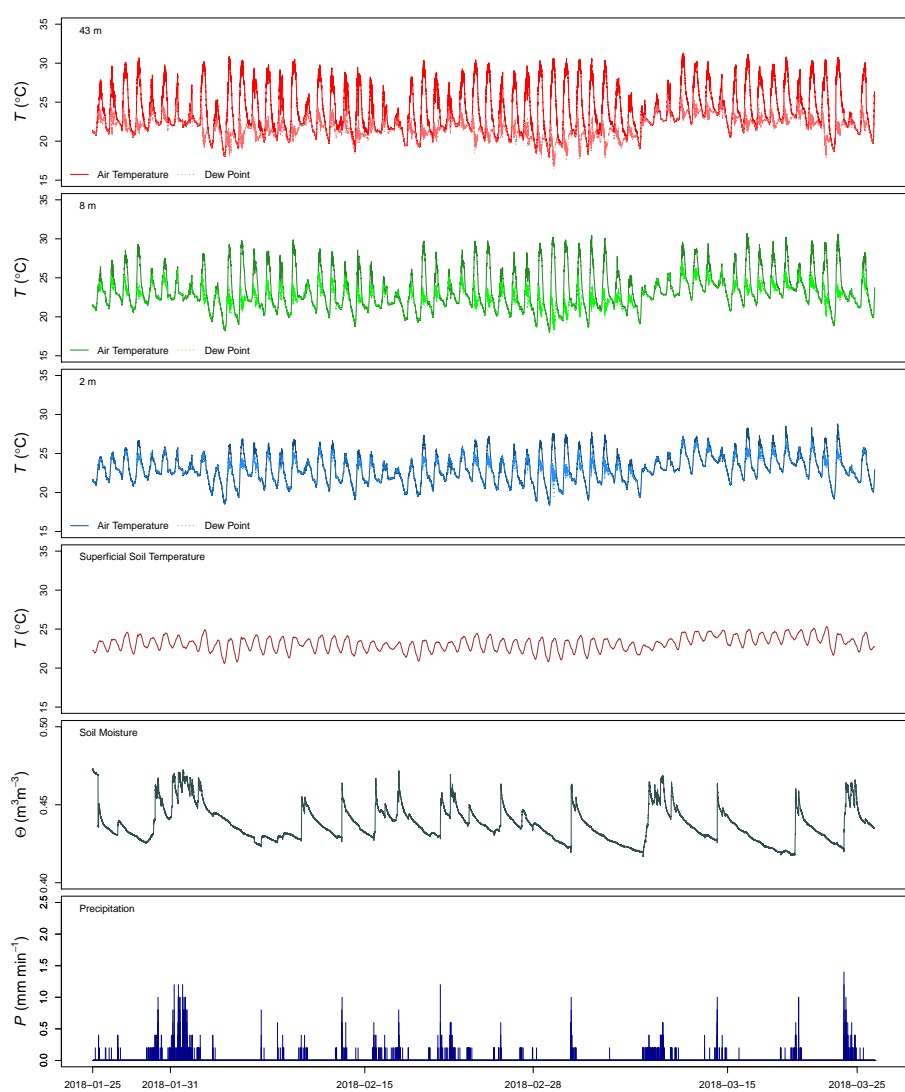

**Figure B1.** Detailed measurements performed at the MRI–plot between 2018-01-24 and 2018-03-26 along the canopy and within the soil.

# Appendix B: Daily variables measured at the MRI–plot

# Appendix C: Boxplots

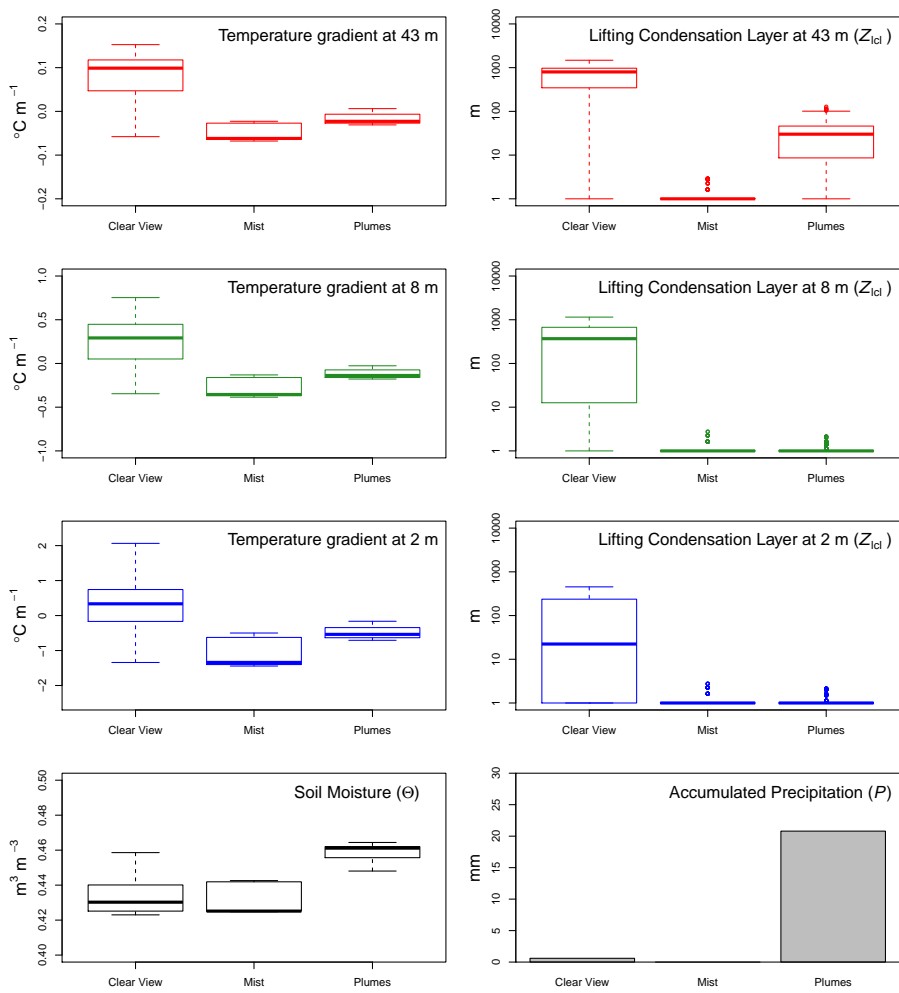

**Figure C1.** Boxplots describing the temperature gradients ($\frac{\Delta \theta_v}{\Delta z}$) and lifting condensation level ($z_{lcl}$) at 43 m, 8 m and 2 m, as well as soil moisture ($\Theta$) and total precipitation ($P$) of the three visual categories evaluated.

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
