# Peer review of "Vapor plumes in a tropical wet forest: spotting the invisible evaporation"

_Hydrology and Earth System Sciences, 2020_

## Referee Comment (RC1) · Shigeki Murakami (Referee) · 3 Apr 2020

Comments to authors

General comments

This paper deals with an interesting phenomenon, visible vapor plumes from wet forest. It is commonly seen in Japan at the time of rainfall and/or just after the cessation of rainfall. It is often observed from my office, and one of my colleagues and I have tried to make a plan to observe it. Even so, we could not come up with the idea and approach to clarify the conditions or mechanisms. Though authors used conventional

instrumentation, the results are clear and reasonable. Now that authors have shown the methodology, we realize how apparent it really is. Anyway, authors succeeded in clarifying the conditions and the mechanism that the plumes are formed, though there are still many unknown processes to elucidate. I appreciate the smart way of observations and analyses. This manuscript is well written and is worth publishing in this journal with minor revision.

Specific comments

1) Page 1, Introduction

There are some researches that show evaporation from forest works as a moisture pump that transports water vapor from the ocean to the inland on a continental scale (Makarieva and Gorshkov, 2007; Makarieva et al., 2013a). Those studies claim that the amount of precipitation in the inland of the continent covered with forest is almost the same with that at the coastal area because of the biotic pump mechanism. Please cite those papers.

2) Page 6, line 5

"The parameters $\Psi S$ and $\Psi L$ were determined using the vapor pressure deficit of the air on each height (Stull, 2016)". $\Psi S$ is calculated using the vapor pressure deficit (even if it is zero), but how is $\Psi L$ estimated? Under the condition of visible vapor plumes the relative humidity (RH) is 100% or more and the vapor pressure deficit is zero. I tried to find the method in Stull (2017; the 2016 version has been revised and seems to be unavailable), but could not.

3) Results and Discussion

RH in the visible vapor plumes is 100% or more, i.e. saturation or supersaturation. However, the plumes continue to grow and evaporation does not stop, because plumes are not stagnant but are moving upward; water vapor along with visible vapor plumes are removed toward the higher altitude due probably to the mechanism proposed by

Makarieva et al. (2013b), i.e. condensation in clouds with drastic reduction in volume of vapor caused by the phase change. At the same time some part of the ambient air is unsaturated, and I think splash droplets keep evaporating because of this unsaturated air. The negative temperature gradient may be caused by the latent heat of vaporization of splash droplet evaporation in the canopy, but it is difficult to know if it is the result of the evaporation or originated from other causes. Authors described visible water vapor plumes only from phenomenological point of view, but the above-mentioned inferred mechanisms that plumes are maintained is worth describing.

4) Page 7, line 19

"because of its timing"; this expression is ambiguous and difficult to follow what it meas. Please add a sentence to explain the detail, like "i.e. mist might be formed early in the morning 201-03-21 and 2018-03-22 but the time lapse video did not work at those times (Table B1)."

5) Page 8, line 28

"there are two sources of aerosols at LSBS". Please add one more source of aerosols. Recent studies proved that a numerous number of bioaerosols are released from forests upon rainfall. For example, Huffman et al. (2013) mentioned in Conclusions, "Our observations indicate that rainfall can trigger intense bursts of bioparticle emission within the forest canopy and massive enhancements of atmospheric bioaerosol concentrations by an order of magnitude or more, from the onset of precipitation through extended periods of high surface wetness after the rainfall (up to one day)." Bioaerosols are integral source of aerosols relevant to rainfall in forest, and please cite paper(s) dealing with this issue at least Huffman et al. (2013).

Technical corrections

6) Page 1, line 10

$\Delta\Theta/\Delta z$ -> $\Delta\theta/\Delta z$

7) Page 1, line 10

Zlcl,43; "Z" is notated in capital letter, but it is in small letter on page 6, line 15 and equation 7. It holds true throughout the manuscript. Please unify the notation.

8) Page 2, line 2

Pease insert "is" between "This" and "because".

9) Page 4, line 9

"TS.5"; I think "TS.0" is correct. Please confirm.

10) Page 4, line 11

"TSS"; There is no description on the definition of TSS. Please clarify.

11) Page 5, Equation 1

"zS"; There is no description on the definition of zS. Please clarify.

12) Page 6, line 2

Please insert "C" between "°" and ")".

13) Page 6, line 4

moist -> saturated

14) Page 7, line 13

Please insert "mm" between "0.2" and "d-1".

15) Page 7, line 18

identify -> identified

16) Page 10, line 16

Spellman (2010) -> (Spellman, 2012)

17) Page 11, line 10

-0.5 °C m-1 -> -1 °C m-1

References

Makarieva, A. M., and V. G. Gorshkov, 2007: Biotic pump of atmospheric moisture as driver of the hydrological cycle on land. Hydrol. Earth Syst. Sci., 11, 1013–1033, doi:10.5194/hess-11-1013-2007.

Makarieva, A. M., V. G. Gorshkov, B.-L. Li. 2013a: Revisiting forest impact on atmospheric water vapor transport and precipitation. Theor. Appl. Climatol., 111, 79–96, doi:10.1007/s00704-012-0643-9.

Makarieva, A. M., V. G. Gorshkov, D. Sheil, A. D. Nobre, and B.-L. Li. 2013b: Where do winds come from? A new theory on how water vapor condensation influences atmospheric pressure and dynamics. Atmos. Chem. Phys., 13, 1039–1056, doi:10.5194/acp-13-1039-2013.

Huffman, J. A., Prenni, A. J., DeMott, P. J., Pöhlker, C., Mason, R. H., Robinson, N. H., Fröhlich-Nowoisky, J., Tobo, Y., Després, V. R., Garcia, E., Gochis, D. J., Harris, E., Müller-Germann, I., Ruzene, C., Schmer, B., Sinha, B., Day, D. A., Andreae, M. O., Jimenez, J. L., Gallagher, M., Kreidenweis, S. M., Bertram, A. K., and Pöschl, U. 2013: High concentrations of biological aerosol particles and ice nuclei during and after rain, Atmos. Chem. Phys., 13, 6151–6164, doi:10.5194/acp-13-6151-2013.

---

## Author Comment (AC1) · 29 Apr 2020

**Reply**
In blue we copied the referee's comments, in black our reply.

**General comments**

This paper deals with an interesting phenomenon, visible vapor plumes from wet forest. It is commonly seen in Japan at the time of rainfall and/or just after the cessation of rainfall. It is often observed from my office, and one of my colleagues and I have tried to make a plan to observe it. Even so, we could not come up with the idea and approach to clarify the conditions or mechanisms. Though authors used conventional instrumentation, the results are clear and reasonable. Now that authors have shown the methodology, we realize how apparent it really is. Anyway, authors succeeded in clarifying the conditions and the mechanism that the plumes are formed, though there are still many unknown processes to elucidate. I appreciate the smart way of observations and analyses. This manuscript is well written and is worth publishing in this journal with minor revision.

**Reply:**
The authors appreciate the time and comments given by the referee. Indeed, this process happens worldwide in different ecosystems and climatic conditions. However, there is few research done about it. We will include all the suggestions and corrections in the revised version of the manuscript.

**Specific comments**
1) Page 1, Introduction
There are some researches that show evaporation from forest works as a moisture pump that transports water vapor from the ocean to the inland on a continental scale (Makarieva and Gorshkov, 2007; Makarieva et al., 2013a). Those studies claim that the amount of precipitation in the inland of the continent covered with forest is almost the same with that at the coastal area because of the biotic pump mechanism. Please cite those papers.

**Reply:**
We appreciate the suggestion of adding the "biotic pump mechanism" in the introduction. We will improve the introduction with the following modifications:

Page 1, Line 17:

"… 2007), where the forest presence at continental scale induced the "biotic pump mechanism" that favored the maintenance of similar precipitation amounts between inland and coastal environments (Makarieva and Gorshkov, 2007; Makarieva et al., 2013a). Meanwhile, the vertical transport …"

Page 1, Line 18:
"… (Trzeciak et al., 2017) that in large ecosystems influence the formation …"

Page

2) Page 6, line 5
"The parameters $\Psi S$ and $\Psi L$ were determined using the vapor pressure deficit of the air on each height (Stull, 2016)". $\Psi S$ is calculated using the vapor pressure deficit (even if it is zero), but how is $\Psi L$ estimated? Under the condition of visible vapor plumes the relative humidity (RH) is 100% or more and the vapor pressure deficit is zero. I tried to find the method in Stull (2017; the 2016 version has been revised and seems to be unavailable), but could not.

**Reply:**

The estimation of both parameters is available in "Section 4.3 Total Water" of Stull (2017). You can check this url to access the document: https://www.eoas.ubc.ca/books/Practical_Meteorology/

It is important to mention that the parameters $\Psi_s$ and $\Psi_L$ are referred in Stull (2017) as $r_s$ and $r_L$. Now that is available the revised version, we will update the references with Stull (2017). Here, it was a description mistake from our side. Instead of using the VPD, we used the saturation and actual vapor pressures to determine both parameters. Consequently, we propose to improve the sentence in page 6, line 5. Also, we will include the required equations to estimate these two parameters. The sentence improvement goes as follow:

"… The parameters $\Psi_s$ and $\Psi_L$ were determined with equations 7 and 8, respectively. These equations requires to know the mass of the liquid water in the air ($m_{liq.air}$), the mass of the dry air ($m_{dry.air}$), the density of the air ($\rho_{air}$) and the density of the liquid water content of the air ($\rho_{LWC}$). These variables were determined using the saturation and actual vapor pressures of the air (Stull, 2017). …"

$$\Psi_s = \frac{m_{liq.water}}{m_{dry.air}} \qquad (7)$$

$$\Psi_L = \frac{\rho_{LWC}}{\rho_{air}} \qquad (8)$$

3) Results and Discussion

RH in the visible vapor plumes is 100% or more, i.e. saturation or super saturation. However, the plumes continue to grow and evaporation does not stop, because plumes are not stagnant but are moving upward; water vapor along with visible vapor plumes are removed toward the higher altitude due probably to the mechanism proposed by Makarieva et al. (2013b), i.e. condensation in clouds with drastic reduction in volume of vapor caused by the phase change. At the same time some part of the ambient air is unsaturated, and I think splash droplets keep evaporating because of this unsaturated air. The negative temperature gradient may be caused by the latent heat of vaporization of splash droplet evaporation in the canopy, but it is difficult to know if it is the result of the evaporation or originated from other causes. Authors described visible water vapor plumes only from phenomenological point of view, but the above-mentioned inferred mechanisms that plumes are maintained is worth describing.

**Reply:**

Thanks for pointing out these mechanisms that complement the "splash droplet evaporation" process during the formation of visible vapor plumes. We will include them in page 8, line 32 as follows:

"… As plumes are not stagnant and continue moving upwards thanks to air convection, the water vapor is removed from the understory towards higher altitudes. The water condensation at the canopy level drastically reduced the volume of water vapor due to the phase change (Makarieva et al., 2013b). This allowed the ambient air to remain unsaturated and keeping the "splash droplet evaporation" process providing continuously more water vapor. "

4) Page 7, line 19

"because of its timing"; this expression is ambiguous and difficult to follow what it meas. Please add a sentence to explain the detail, like "i.e. mist might be formed early in the morning 2018-03-21 and 2018-03-22 but the time lapse video did not work at those times (Table B1)."

**Reply:**

Thanks for pointing out this phrase. Following your recommendation we will add the following sentence in page 7, line 19:

"…7:00 a.m. Mist might be formed early in the morning during the sampling dates 2018-03-21 and 2018-03-22. However, the time lapse video did not work at those times (Table B1). These mist …"

5) Page 8, line 28
"there are two sources of aerosols at LSBS". Please add one more source of aerosols. Recent studies proved that a numerous number of bioaerosols are released from forests upon rainfall. For example, Huffman et al. (2013) mentioned in Conclusions, "Our observations indicate that rainfall can trigger intense bursts of bioparticle emission within the forest canopy and massive enhancements of atmospheric bioaerosol concentrations by an order of magnitude or more, from the onset of precipitation through extended periods of high surface wetness after the rainfall (up to one day)." Bioaerosols are integral source of aerosols relevant to rainfall in forest, and please cite paper(s) dealing with this issue at least Huffman et al. (2013).
**Reply:**
Thanks for this suggestion. We will improve the discussion with the following improvements and additions:

Page 8, line 28:
"… there are different sources of …"

Page 8, line 29:
"… Loescher et al., 2004). A second"

Page 8, line 30:
"… LSBS. Additionally, it cannot be discarded the presence of bioparticles (e.g., airborne bacteria, fungi, pollen, plant fragments, organic compounds) as a source of aerosols from the forests (Huffman et al., 2013, Pöschlet al., 2010, Valsan et al., 2015). The high intensity rains may induce the bioparticles burst from the forest canopy. These bioparticles have been in Australia (Bigg et al., 2015), India (Valsan et al., 2015), Mexico (Rodriguez-Gomez et al., 2020), and the Amazon (Pöschlet al., 2010). Also, convective rains transport from the free troposphere into the boundary layer a portion of the required aerosols …"

Technical corrections

6) Page 1, line 10
ΔΘ/Δz -> Δθ/Δz
**Reply:**
Thanks, we will change it.

7) Page 1, line 10
Zlcl,43; "Z" is notated in capital letter, but it is in small letter on page 6, line 15 and equation 7. It holds true throughout the manuscript. Please unify the notation.
**Reply:**
Thanks for making note of this. We will unify the notation with "z".

8) Page 2, line 2 Pease insert "is" between "This" and "because".
**Reply:**

Thanks. We will include it.

9) Page 4, line 9 "TS.5"; I think "TS.0" is correct. Please confirm.
**Reply:**
The variable $T_{s.5}$ is correct. This section described the estimation of superficial soil temperature ($T_{s.0}$) because the soil data from the field was collected at 5 cm depth.

10) Page 4, line 11 "TSS"; There is no description on the definition of TSS. Please clarify.
**Reply:**
Thanks for making note of this. This should be $T_{s.0}$, so we will change it in the revised version of the manuscript.

11) Page 5, Equation 1 "zS"; There is no description on the definition of zS. Please clarify.
**Reply:**
Thanks. This is a typo in the manuscript. It should be $z_b$, and we will change it.

12) Page 6, line 2 Please insert "C" between "∘ " and ")".
**Reply:**
Thanks. We will change it.

13) Page 6, line 4 moist -> saturated
**Reply:**
Thanks. We will change it.

14) Page 7, line 13 Please insert "mm" between "0.2" and "d-1".
**Reply:**
Thanks. We will add it.

15) Page 7, line 18 identify -> identified
**Reply:**
Thanks. We will change it.

16) Page 10, line 16 Spellman (2010) -> (Spellman, 2012)
**Reply:**
Thanks. We will change it.

17) Page 11, line 10 -0.5 ∘C m-1 -> -1 ∘C m-1
**Reply:**
Thanks. We will change it.

**References**

Bigg, E. K., Soubeyrand, S., and Morris, C. E.: Persistent after-effects of heavy rain on concentrations of ice nuclei and rainfall suggest abiological cause, Atmospheric Chemistry and Physics, 15, 2313–2326, https://doi.org/10.5194/acp-15-2313-2015, 2015.

Makarieva, A. M., and V. G. Gorshkov, 2007: Biotic pump of atmospheric moisture as driver of the hydrological cycle on land. Hydrol. Earth Syst. Sci., 11, 1013–1033, doi:10.5194/hess-11-1013-2007.

Makarieva, A. M., V. G. Gorshkov, B.-L. Li. 2013a: Revisiting forest impact on atmospheric water vapor transport and precipitation. Theor. Appl. Climatol., 111, 79–96, doi:10.1007/s00704-012-0643-9.

Makarieva, A. M., V. G. Gorshkov, D. Sheil, A. D. Nobre, and B.-L. Li. 2013b: Where do winds come from? A new theory on how water vapor condensation influences atmospheric pressure and dynamics. Atmos. Chem. Phys., 13, 1039–1056, doi:10.5194/acp-13-1039-2013.

Huffman, J. A., Prenni, A. J., DeMott, P. J., Pöhlker, C., Mason, R. H., Robinson, N. H., Fröhlich-Nowoisky, J., Tobo, Y., Després, V. R., Garcia, E., Gochis, D. J., Harris, E., Müller-Germann, I., Ruzene, C., Schmer, B., Sinha, B., Day, D. A., Andreae, M. O., Jimenez, J. L., Gallagher, M., Kreidenweis, S. M., Bertram, A. K., and Pöschl, U. 2013: High concentrations of biological aerosol particles and ice nuclei during and after rain, Atmos. Chem. Phys., 13, 6151–6164, doi:10.5194/acp-13-6151-2013.

Pöschl, U., Martin, S. T., Sinha, B., Chen, Q., Gunthe, S. S., Huffman, J. A., Borrmann, S., Farmer, D. K., Garland, R. M., Helas, G., Jimenez,J. L., King, S. M., Manzi, A., Mikhailov, E., Pauliquevis, T., Petters, M. D., Prenni, A. J., Roldin, P., Rose, D., Schneider, J., Su, H., Zorn, S. R., Artaxo, P., and Andreae, M. O.: Rainforest Aerosols as Biogenic Nuclei of Clouds and Precipitation in the Amazon, Science, 329,1513–1516, https://doi.org/10.1126/science.1191056, 2010.

Rodriguez-Gomez, C., Ramirez-Romero, C., Cordoba, F., Raga, G. B., Salinas, E., Martinez, L., Rosas, I., Quintana, E. T., Maldonado, L. A.,Rosas, D., Amador, T., Alvarez, H., and Ladino, L. A.: Characterization of culturable airborne microorganisms in the Yucatan Peninsula,Atmospheric Environment, 223, 117 183, https://doi.org/10.1016/j.atmosenv.2019.117183, 2020

Valsan, A. E., Priyamvada, H., Ravikrishna, R., Després, V. R., Biju, C., Sahu, L. K., Kumar, A., Verma, R., Philip, L., and Gunthe, S. S.: Mor-phological characteristics of bioaerosols from contrasting locations in southern tropical India – A case study, Atmospheric Environment,122, 321–331, https://doi.org/10.1016/j.atmosenv.2015.09.071, 2015.

---

## Referee Comment (RC2) · Shigeki Murakami (Referee) · 1 May 2020

Comments to authors 2

General comments 2
The manuscript is revised perfectly except for "Specific comments, 2)"
that is needed additional modification and explanation.
I noticed additional typos and listed them in Technical corrections 2.

Specific comments 2
2) Page 6, lines 4 to 8 of the original manuscript
As authors mentioned in their reply, $\Psi_S$ is defined as mass of water
vapor divided by mass of dry air. The numerator in the new equation
(7) in the reply is not $m_{liq.water}$ but $m_{vap.liq}$ as shown below.

$$\Psi_S = \frac{m_{vap.water}}{m_{dry.air}} \qquad (7)$$

The new equation (8) is rewritten as the equation (8').

$$\Psi_L = \frac{\rho_{LWC}}{\rho_{air}} \qquad (8)$$

$$\Psi_L = \frac{m_{liq.water}}{m_{dry.air}} \qquad (8')$$

The problem is how the value of $\rho_{LWC}$ in the equation (8) or $m_{liq.water}$ in
(8') is estimated. Generally speaking, direct measurement of liquid
water content (LWC) in the air is difficult and it seems that no
instrument to measure LWC was installed at the site. I think authors
used some estimated values. Please specify how $\rho_{LWC}$ or $m_{liq.water}$ was
estimated.

Technical corrections 2
18) Page 2, line 23
Please insert "being" between "techniques" and "able".

19) Page 7, line 20

Please add "with" between "2ºC" and "respect".

20) Page 8, line 27
a lower density than air above -> the dew point temperature

21) Page 8, line 31
formation of -> they form

22) Page 10, line 7
is more easy to move upwards the parcels of air
 -> it is easier for the air parcels to move upwards

23) Page 10, line 18
Spellman (2012) -> (Spellman, 2012)

---

## Author Comment (AC2) · 25 May 2020

**Reply**
In blue we copied the referee's comments, in black our reply.

Comments to authors 2
General comments 2
The manuscript is revised perfectly except for "Specific comments, 2)" that is needed additional modification and explanation. I noticed additional typos and listed them in Technical corrections 2.

Specific comments 2
2) Page 6, lines 4 to 8 of the original manuscript. As authors mentioned in their reply, ΨS is defined as mass of water vapor divided by mass of dry air. The numerator in the new equation (7) in the reply is not mliq.water but mvap.liq as shown below.

$$\Psi_S = \frac{m_{vap.water}}{m_{dry.air}} \qquad (7)$$

The new equation (8) is rewritten as the equation (8').
$$\Psi_L = \frac{\rho_{LWC}}{\rho_{air}} \qquad (8)$$

$$\Psi_L = \frac{m_{liq.water}}{m_{dry.air}} \qquad (8')$$

**Reply:**
Thanks for the correction of equations 7 and 8. Actually, we meant the vapor water in the air. We will include those changes in the final version of the manuscript.

The problem is how the value of $\rho_{LWC}$ in the equation (8) or $m_{liq.water}$ in (8') is estimated. Generally speaking, direct measurement of liquid water content (LWC) in the air is difficult and it seems that no instrument to measure LWC was installed at the site. I think authors used some estimated values. Please specify how $\rho_{LWC}$ or $m_{liq.water}$ was estimated.

**Reply:**
Indeed, we used an estimation of the liquid water content of the air according with Thomsom (2007). Due to the characteristics of the vapor plumes formed during the monitoring period, we choose the liquid water content of fog events of 0.05 g m$^{-3}$. We proposed to add the following sentences in the methodology as follows:

"Due to the lack of instrumentation to estimate the liquid water content in the air, we used a fixed value of 0.05 g m$^{-3}$. This value corresponds to the liquid water in the air reported by Thompson (2007) for fog events. Using this value and the specific humidity of the air…..The selection of this value was based on (1) the similarity in terms of color and transparency between the vapor plumes and fog, and (2) because both types of events occur close to the ground surface."

Technical corrections 2
18) Page 2, line 23
Please insert "being" between "techniques" and "able".
**Reply:**
Thanks for the correction. We will include it in the final manuscript.

19) Page 7, line 20
Please add "with" between "2°C" and "respect".
**Reply:**
Thanks for the correction. We will include it in the final manuscript.

20) Page 8, line 27
a lower density than air above -> the dew point temperature
**Reply:**
Thanks for the correction. We will include it in the final manuscript.

21) Page 8, line 31
formation of -> they form
**Reply:**
Thanks for the correction. We will include it in the final manuscript.

22) Page 10, line 7
is more easy to move upwards the parcels of air
-> it is easier for the air parcels to move upwards
**Reply:**
Thanks for the suggestion. We will include it in the final manuscript.

23) Page 10, line 18
Spellman (2012) -> (Spellman, 2012)
**Reply:**
Thanks for the correction. We will include it in the final manuscript.

**References**

Thompson, A.: Simulating the adiabatic ascent of atmospheric air parcels using the cloud chamber, Department of Meteorology, Penn State, 2007.

---

## Referee Comment (RC3) · Shigeki Murakami (Referee) · 27 May 2020

The manuscript has been impeccably revised. I think it is acceptable as it is.

---

## Referee Comment (RC4) · Anonymous Referee #2 · 25 Jun 2020

This paper attempts to describe the formation process of vapor plumes in a tropical forest environment due to evaporation processes happening during rain events. This phenomenon is quite interesting and probably contributes to water cycling in tropical wet forests. However, I do have substantial doubts about the scientific quality and the overall rationale of the study. The various reasons are outlined below:

**1. Scientific objectives / rationale:** To my opinion, the scientific objective and the conclusions about the identification of visible vapor plumes in a Tropical Wet Forest are relatively weak and the measurement setup is not suitable to derive reliable evaporation estimates during these events. If any observer (or any camera) can identify the
vapour plumes above the tropical forest what is the novelty of this paper? What is the contribution to the scientific knowledge about water cycling in the tropics? One scientific objective could have been to estimate the contribution of these vapour plumes to the total annual evapotranspiration (ET) flux in these environments. To achieve this, more sophisticated measurements (e.g., 3D wind components) and a detailed literature study comparing the magnitude of evaporation fluxes from vapor plumes at this site in Costa Rica to ET fluxes measured by eddy covariance at other comparable tropical sites (e.g. Puerto Rico, Amazon Basin) should have been performed (Holwerda et al., 2012; Paca et al., 2019). In addition, a modelling exercise using remote sensing data (i.e. land surface temperature) would probably have been feasible. However, the authors did not attempt to investigate the role of these plumes for the hydrological cycle.

**2. Experimental setup:** According to the paper, the vertical air temperature profiles using the Hobo sensors were not actively aspirated, which, however, should be the case to achieve sufficient accuracy when measuring vertical air temperature profiles. Additionally, wind speed measurements were apparently entirely lacking (or were made at 10m height only, according to Jiménez-Rodríguez et al., 2020). For this type of analysis, a vertical wind speed profile within and above the canopy or 3D wind components should be measured.

**3. Methodology:** According to my understanding, the method used to calculate the evaporation flux within and above the forest canopy violates fundamental micrometeorological theory. The energy balance (EB) equation was used to derive evaporation within and above the canopy according to Jiménez-Rodríguez et al., 2020, although none of the EB components were directly measured in the field. The term in the energy balance equation refers to evapotranspiration, which also includes transpiration. How can it be justified that transpiration was indeed zero under these conditions? The calculation of net radiation (Rn) components within the canopy is quite complex due to multiple extinction processes. Applying the equations give in the appendix of Jiménez-

Rodríguez et al., 2020 does not seem to be reasonable to derive an in-canopy profile of Rn. Furthermore, the sensible heat flux within and above the canopy was calculated by applying flux-profile relationships, which involves estimation of the aerodynamic resistance. Flux-profile relationships are typically expressed by formulations based on the Monin-Obukhov similarity (MOST) theory within the lowest 10% of the atmospheric boundary layer where exchange fluxes are considered to be constant with height. According to MOST, the aerodynamic resistance can only be calculated in cases when a logarithmic profile of the horizontal wind speed is present (Thom, 1975; Foken, 2017) - typically some meters above the canopy. Hence, the equations applied inside the canopy are invalid. Moreover, wind speed measurements are required to derive the aerodynamic resistance. As atmospheric turbulence is random, the use of parameterizations based on ancient wind speed measurements is not feasible. Due to these fundamental aspects, the numbers provided in the paper are only a crude approximation and do not provide a basis to derive a solid scientific conclusion.

**References:**

Foken, T. (2017) Micrometerology, Springer, Berlin, Heidelberg.

Holwerda, F., Bruijnzeel, L.A., Scatena, F.N., et al. (2012) Wet canopy evaporation from a Puerto Rican lower montane rain forest: The importance of realistically estimated aerodynamic conductance. Journal of Hydrology 414, 1-15, doi: 10.1016/j.jhydrol.2011.07.033.

Jiménez-Rodríguez, C. D., Coenders-Gerrits, M., Wenninger, J., Gonzalez-Angarita, A., and Savenije, H.: Contribution of understory evaporation in a tropical wet forest during the dry season, Hydrol. Earth Syst. Sci., 24, 2179–2206, https://doi.org/10.5194/hess-24-2179-2020, 2020.

Paca, V.H.M., Espinoza-Dávalos, G.E., Hessels, T.M., et al. (2019) The spatial variability of actual evapotranspiration across the Amazon River Basin based on remote sensing products validated with flux towers. Ecological Processes 8(1), 6, doi: 10.1186/s13717-019-0158-8.

Thom, A.S. (1975) Vegetation and the Atmosphere. Monteith, J.L. (ed), pp. 57–109,

Academic Press, London.

---

## Author Comment (AC3) · 14 Jul 2020

**Reply**

We thank the reviewer for his/her time to provide us feedback on our study. We copied the blue the referee's comment and in black our reply.

This paper attempts to describe the formation process of vapor plumes in a tropical forest environment due to evaporation processes happening during rain events. This phenomenon is quite interesting and probably contributes to water cycling in tropical wet forests. However, I do have substantial doubts about the scientific quality and the overall rationale of the study. The various reasons are outlined below:

1. Scientific objectives / rationale: To my opinion, the scientific objective and the conclusions about the identification of visible vapor plumes in a Tropical Wet Forest are relatively weak and the measurement setup is not suitable to derive reliable evaporation estimates during these events. If any observer (or any camera) can identify the vapour plumes above the tropical forest what is the novelty of this paper? What is the contribution to the scientific knowledge about water cycling in the tropics? One scientific objective could have been to estimate the contribution of these vapour plumes to the total annual evapotranspiration (ET) flux in these environments. To achieve this, more sophisticated measurements (e.g., 3D wind components) and a detailed literature study comparing the magnitude of evaporation fluxes from vapor plumes at this site in Costa Rica to ET fluxes measured by eddy covariance at other comparable tropical sites (e.g. Puerto Rico, Amazon Basin) should have been performed (Holwerda et al., 2012; Paca et al., 2019). In addition, a modelling exercise using remote sensing data (i.e. land surface temperature) would probably have been feasible. However, the authors did not attempt to investigate the role of these plumes for the hydrological cycle.

**Reply:**

We understand there is some confusion on the objective of our study. Our objective is not to quantify the contribution of vapor plumes to total evaporation. This would -as the reviewer correctly mentions- require a totally different setup. The objective of our study is more preliminary and is twofold: 1) to identify vapor plumes and 2) to explain when and why these plumes occur using meteorological data.

The formation of visible vapor plumes is commonly known to happen after rain events (Page 2, line 21), but the specific conditions of how they form, has not been scientifically described yet. A detailed description of when this commonly known phenomenon occurs, and how it links to local hydrology at site level will help to understand different offsets in other processes such as interception of precipitation. Moreover, there is a lack of techniques that are able "to characterize the occurrence of these plumes close to the surface" (Page 2, line 22), and this manuscript provides a procedure that may be used to recognize the meteorological conditions needed for the formation of these visible vapor plumes.

We think understanding this is a pre-requisite before we are able to quantify such a complex process as vapor plumes. Additionally, although we would have loved to quantify the vapor contribution to total evaporation, the difficulty lies into the circumstances vapor plumes form. Visible vapor plumes that we studied, happen during rainy days and under these conditions (continuous rain events) the performance of more sophisticated instruments is compromised (Centre for Atmospheric Science, 2020). Instruments such as sonic anemometers (e.g., CSAT3, CSAT3B) and Open Path $CO_2$/$H_2O$ Analyzers (e.g., LI-7500) are strongly affected by high humidity and rainfall (Campbell Scientific Inc., 2017, 2019, Foken et al., 2012b, LI-COR, 2016, Moncrieff et al., 2005). The presence of rain causes departures from the measurements increasing the sonic speed (Camuffo 2019, Kelton and Bricout, 1964, Peters et al., 1998) or blocking the face of the transducers (Campbell Scientific Inc., 207, 2017) causing a frequency loss during rain events (Zhang et al., 2016). These conditions trigger the need to screen the data and filter the rain events to remove poor-quality data (Mauder and Zeeman, 2018) requiring complex gap filling procedures afterwards (Moncrieff et al.,2005).

Also using remote sensing data would not be feasible since land surface temperature products have a too coarse spatial (e.g., Landsat: 60 m, MODIS: ~1 km, Aster: 90 m) and temporal resolution (e.g., Landsat: 16 days, MODIS: twice daily, Aster: twice daily). While vapor plumes occur in time windows that last minutes and cover small areas around the forest.

To the contrary, conventional measuring devices for temperature and relative humidity (e.g., HOBO sensors) are reliable when installed with multi-plate plastic shelters, registering a deviation of just 1.6 % from the mean temperature of the fully ventilated sensors under tropical conditions during sunny days (da Cunha, 2015). Meanwhile, during rainy days and night-time conditions, conventional air temperature sensors have low biases when compare against active ventilated sensors (Terando et al., 2017). Also, measuring devices based on 3D wind components (e.g., eddy-covariance systems) are developed to measure water in gas form (Foken et al., 2012a) and are not intended to measure vapor plumes that are ascending clusters of tiny water particles (Spellman, 2012). Therefore, we decided to use conventional meteorological instruments to study vapor plumes.

To clarify to the reader why we did not quantify the contribution of vapor plumes, we will add the reasoning, as described above. Consequently, we proposed the following changes to the manuscript follows:

Add to the introduction in page 2, line 23:
"… to the surface. Visible vapor plumes are classified as ascending clouds formed by clusters of tiny particles of water in liquid form (Spellman, 2012). This characteristic makes difficult to measure them with sophisticated systems based on 3D wind components (e.g., eddy-covariance systems) that are developed to measure water in gas form (Foken et al., 2012a). This type of measurements are sensitive to rainy and high humidity conditions (Camuffo 2019, Foken et al., 2012b, Kelton and Bricout, 1964, Moncrieff et al., 2005, Mauder and Zeeman, 2018, Peters et al., 1998) making difficult to use them to identify the occurrence of visible vapor plumes in forested ecosystems. This mismatch between measurement systems and target phenomena, underlines the need to identify the conditions under which visible vapor plumes are formed. This type of methodological constraints requires an innovative data analysis approach, which is the focus of this paper."

Improve the manuscript aims as follows:

"This work aims (1) to test an innovative approach to link visual information and conventional meteorological data describing a local hydrological phenomenon. Also, (2) to identify the meteorological conditions when visible vapor plumes are present in a Tropical Wet Forest, and it tries (3) to explain the processes involve on the formation of these visible vapor plumes. The data analysis is based on conventional meteorological data vertically distributed along the forest canopy layer and time-lapse videos during day-time conditions."

Add in Page 3, line 19:

"… H21-USB). Meteorological data collected along the tower and soil temperature data were recorded with 1 min and 5 min averages, respectively. All data was summarized in 5 min time intervals for the analysis. A Bushnell® … "

2. Experimental setup: According to the paper, the vertical air temperature profiles using the Hobo sensors were not actively aspirated, which, however, should be the case to achieve sufficient accuracy when measuring vertical air temperature profiles.

**Reply:**
It is true that the sensors were not actively aspirated. However, the use of the radiation shields together with HOBO air temperature sensors allows keeping the mean absolute error during day time below 0.3 ºC (da Cunha, 2015, Terando et al., 2017) in warm tropical ecosystems. In addition, the measurement of minimum or night-time temperatures does not require the cover of the radiation shield to keep low biases on the mean temperature (<0.5 ºC) due to the reduced or total absence of solar radiation (Terando et al., 2017). Also, the shelter provided by the forest canopy for the measurements carried out at 2 m helps to record similar temperatures to the surrounding near-surface environment (Lundquist & Huggett, 2008).

We added this explanation to the manuscript in Page 3, line 14 as follows:
"The use of radiation shields together with conventional air temperature sensors allows keeping a mean absolute error during day time in warm tropical ecosystems below 0.3 ºC (da Cunha, 2015, Terando et al., 2017). Also, the shelter provided by the forest canopy for the measurements carried out at 2 m helps to record similar temperatures to the surrounding near-surface environment (Lundquist & Huggett, 2008). Meanwhile, the measurement of minimum air temperatures or night-time temperatures does not require the cover of the radiation shield to keep low biases (<0.5 ºC) on the mean air temperature due to the reduced or total absence of solar radiation (Terando et al., 2017)."

Additionally, wind speed measurements were apparently entirely lacking (or were made at 10m height only, according to Jiménez-Rodríguez et al., 2020). For this type of analysis, a vertical wind speed profile within and above the canopy or 3D wind components should be measured.

**Reply:**
We agree that measurements of a 3D sonic anemometer would have been better under normal circumstances. However, we study the temperature gradients under humid conditions and highly rainy conditions, where sonic anemometers often have their limitations (Camuffo 2019, Foken et al., 2012b, Kelton and Bricout, 1964, Moncrieff et al., 2005, Mauder and Zeeman, 2018, Peters et al., 1998). Also, the main analysis is based on temperature gradients, where vertical wind speed
measurements are not required as the reviewer mentions.

The authors propose to improve the following sentences to clarify this:

Add in Page 6, line 10:

"… gradient $\left(\frac{\Delta \theta}{\Delta z}\right)$ due the absence of wind profile measurements to determine the atmospheric stability parameter along the tower. Values of …"

Add the reference (Stull, 2017) in page 6, line 11.

And specify some details of data collection adding this in Page 3, line 19:

"… H21-USB). Meteorological data collected along the tower and soil temperature data were recorded with 1 min and 5 min averages, respectively. All data was summarized in 5 min time intervals for the analysis. A Bushnell® … "

3. Methodology: According to my understanding, the method used to calculate the evaporation flux within and above the forest canopy violates fundamental micrometeorological theory. The energy balance (EB) equation was used to derive evaporation within and above the canopy according to Jiménez-Rodríguez et al., 2020, although none of the EB components
were directly measured in the field.
**Reply:**
The method used to calculate evaporation is based on the Energy Balance Method as it is described in detail by Jiménez-Rodríguez *et al*. (2020). It was not possible to measure directly the different components of the energy balance method due to constraints of instrumentation availability for a trial study. However, our aim was also not to exactly quantify
evaporation. We just used the estimated evaporation by Jiménez-Rodríguez *et al*. (2020) as a proxy.

Aiming to prevent any confusion between the manuscript objectives and the source of evaporation data, we propose the following:

Eliminate equation 8 and the information contained in page 6 between the lines 18 and 23. Adding the following text instead:

"An estimation of the evaporation during the monitored period was retrieved from Jiménez-Rodríguez *et al.* (2020). This estimation was just used as a proxy and is based on the vertical transport and neglects the advected energy on the forest canopy and is used as a reference of the evaporation process during the monitoring period."

Change the caption of table 1 as follows:

"Table 1. Daily summary of precipitation and evaporation at 43 m, 8 m and 2 m height according to Jiménez-Rodríguez *et al.* (2020) for the experimental site during the monitoring period."

Modify the caption of Figure 3 as follows:

"Figure 3. Virtual potential temperature ($\vartheta_v$), lifting condensation level ($z_{lcl}$) in an untransformed semi–logarithmic scale, and temperature gradient $\left(\frac{\Delta\theta}{\Delta z}\right)$ at 43 m, 8 m and 2 m height. Additionally, precipitation (P) and soil moisture (Θ) are also shown during the visual monitoring between 2018-03-21 and 2018-03-25. Evaporation ($E$) measurements were retrieved from Jimenez-Rodriguez, *et al.* (2020). Background colored areas denoted the three categories in which the photographs were classified: Clear View, Mist and Plumes."

Add the reference (Jiménez-Rodríguez *et al.,* 2020) in page 7, line 11.

The term in the energy balance equation refers to evapotranspiration, which also includes transpiration. How can it be justified that transpiration was indeed zero under these conditions?
**Reply:**
The authors consider important to highlight the evaporation definition used in this manuscript (Page 1, line 13). We used the term forest evaporation as the "*mixture of water vapor originated from water intercepted on plant surfaces, soil water and plant transpiration (Roberts, 1999; Savenije, 2004; Shuttleworth, 1993)*". Consequently, we included all water vapor sources that originated the measured evaporation. Also, the authors never considered the transpiration to be zero, instead we mentioned that transpiration "*is likely reduced by the low vapor pressure deficit*" (Page 11, line 5) but not stopped (this
manuscript). Also, the evaporation data was retrieved from Jiménez-Rodríguez *et al.* (2020) who mentioned that 44.2 % of the evaporated precipitation corresponds to canopy transpiration and the accumulation of 29.4% of the leaf area index increases the potential sources of transpiration. Moreover, we propose that the main source of water vapor feeding the formation of vapor plumes is linked to the evaporation of intercepted water in plant surfaces on the canopy depressions due to two reasons:
1. the splash droplet evaporation process does not need extra energy to occur,
    2. the low leaf area index of the canopy depressions can be translated into a reduced contribution of transpiration to the formation of vapor plumes,
    3. the canopy depressions allow a free vertical path for the water vapor to ascend and from the buoyant cloud after reducing its potential air temperature.
The authors propose to add the following to clarify the source of water vapor for the formation of vapor plumes:

Add in Page 8, line 1:

"… soil moisture. These days were characterized by cumulus clouds crossing the sky above the forest canopy in day time. Any water vapor ascending from the forest canopy will need to reach a height of more than 100 m to form visible vapor plumes (Figure 3). Also, on …"

Add in Page 8, line 26:

"The visible vapor plumes can be spotted on the canopy depressions surrounding the tower (Figure 1). These depressions are characterized by a low leaf area index and shorter canopy height, which translates into areas with
low potential to produce transpiration during rain events. This implies that the main source of water vapor is linked to water evaporated from wet surfaces and soil evaporation, while transpiration may contributes to a lesser extent."

Add in Page 8, line 32:

"… As plumes are not stagnant and continue moving upwards thanks to air convection, the water vapor is removed from the understory towards higher altitudes. The water condensation at the canopy level drastically reduced the volume of water vapor due to the phase change (Makarieva et al., 2013). This allowed the ambient air to remain unsaturated and keeping the "splash droplet evaporation" process providing continuously more water vapor. "

The calculation of net radiation (Rn) components within the canopy is quite complex due to multiple extinction processes. Applying the equations give in the appendix of Jiménez-Rodríguez et al., 2020 does not seem to be reasonable to derive an in-canopy profile of Rn. Furthermore, the sensible heat flux within and above the canopy was calculated by applying fluxprofile relationships, which involves estimation of the aerodynamic resistance. Flux-profile relationships are typically expressed by formulations based on-the Monin-Obukhov similarity (MOST) theory within the lowest 10% of the atmospheric boundary layer where exchange fluxes are considered to be constant with height. According to MOST, the aerodynamic resistance can only be calculated in cases when a logarithmic profile of the horizontal wind speed is present (Thom, 1975; Foken, 2017) - typically some meters above the canopy. Hence, the equations applied inside the canopy are invalid. Moreover, wind speed measurements are required to derive the aerodynamic resistance. As atmospheric turbulence is random, the use of parameterizations based on ancient wind speed measurements is not feasible. Due to these fundamental aspects, the numbers provided in the paper are only a crude approximation and do not provide a basis to derive a solid scientific conclusion.

**Reply:**

We agree that our evaporation estimates are not precise. As said before, we use the evaporation estimates just as a proxy. However, our main analysis focuses on the vapor plumes formation. This process is based on the virtual potential air temperature ($\Theta_v$), lifting condensation level ($Z_{lcl}$), and temperature gradient $\left(\frac{\Delta\theta_v}{\Delta z}\right)$ that are estimated based on the continuous measurements of air temperature, soil temperature, and air humidity. None of these variables were estimated from old data or based on approximations but were all measured at the site during the study period. Also, the estimation of these variables (equations 1 to 7) does not depend on the aerodynamic resistance. This issue of the aerodynamic resistance applies only to the estimation of evaporation carried out in other paper used as a source of information (Page 6, line 21). We clarified in the manuscript that evaporation is only used as an indication and that our analysis focused on the temperature gradients that are not making use of MOST-assumptions. It is important to acknowledge the need for more studies focusing on this process now that the main elements of the formation of visible vapor plumes were identified in this paper. These studies should focus on (1) the quantification of the water vapor linked to the visible plumes, (2) the frequency of occurrence in a yearly basis in tropical and non-tropical regions, and (3) improve the identification procedure of visible vapor plumes through more detailed monitoring processes. Consequently, we recommend to:

Add in Page 10, line 20:

[revised manuscript text omitted]

---

## Author Comment (AC4) · 14 Jul 2020

All the proposed changes in the previous replies will be incorporated in the improved manuscript.

---

## Author Response (AR1)

**Reply to Editor Comments**

In blue we copied the comments of the editor, in black our replies.

Dear authors,

following up on the reviews received for your contribution on 'Vapor plumes in a tropical wet forest - spotting the invisible evaporation', I invite you to submit a revised version of your manuscript. Given the fact that the two reviews are exposing rather contrasting viewpoints - please pay particular attention in the revised version to address issues obviously related to the potential for misunderstandings on the objectives of your work (e.g., stating clearly that it is not targeting a quantification of the contribution of vapour plumes to total evaporation, but rather exposing preliminary results on an approach for identifying the meteorological conditions triggering visible vapor plumes).

**Reply:**

We improved the work objectives as follows (Page 3, line 6):

"This work aims (1) to test an innovative approach to link visual information and conventional meteorological data describing a local hydrological phenomenon. Also, (2) to identify the meteorological conditions when visible vapor plumes are present in a Tropical Wet Forest, and it tries (3) to explain the processes involve on the formation of these visible vapor plumes. The data analysis is based on conventional meteorological data vertically distributed along the forest canopy layer and time-lapse videos during day-time conditions."

Along similar lines, please include in the manuscript further details on the design of the proposed experimental set-up and appropriateness for measuring vertical air temperature profiles, as well as wind speed measurements.

**Reply:**

Add to the introduction in page 2, line 35:

"… to the surface. Visible vapor plumes are classified as ascending clouds formed by clusters of tiny particles of water in liquid form (Spellman, 2012). This characteristic makes difficult to measure them with sophisticated systems based on 3D wind components (e.g., eddy-covariance systems) that are developed to measure water in gas form (Foken et al., 2012a). This type of measurements are sensitive to rainy and high humidity conditions (Camuffo 2019, Foken et al., 2012b, Kelton and Bricout, 1964, Moncrieff et al., 2005, Mauder and Zeeman, 2018, Peters et al., 1998) making difficult to use them to identify the occurrence of visible vapor plumes in forested ecosystems. This mismatch between measurement systems and target phenomena, underlines the need to identify the conditions under which visible vapor plumes are formed. This type of methodological constraints requires an innovative data analysis approach, which is the focus of this paper."

We added this explanation to the manuscript in Page 4, line 4 as follows:

"The use of radiation shields together with conventional air temperature sensors allows keeping a mean absolute error during day time in warm tropical ecosystems below 0.3 °C (da Cunha, 2015, Terando et al., 2017). Also, the shelter provided by the forest canopy for the measurements carried out at 2 m helps to record similar temperatures to the surrounding near-surface environment (Lundquist & Huggett, 2008). Meanwhile, the measurement of minimum air temperatures or night-time temperatures does not require the cover of the radiation shield to keep low biases (<0.5 °C) on the mean air temperature due to the reduced or total absence of solar radiation (Terando et al., 2017)."

We also proposed an experimental setup for measuring the plumes at the end of the Results and Discussion section. This methodology opens the opportunity to use available data sets collected worldwide (Page 13, line 1):

50

"Further studies aiming to analyze in more detail the occurrence of visible vapor plumes and its contribution to the hydrological cycle will need to considerer the conditions that give origin to this phenomenon: air convection, precipitation presence, and lifting condensation level at the top of the canopy. The identification of air convection should be based on direct measurements of ground heat flux

55

(e.g, soil heat flux plates), a more detailed air temperature profile along de forest canopy (e.g, using distributed temperature sensing), and multiple wind speed measurements along the canopy profile (e.g, at least one per canopy layer plus one above the canopy). This set of measurements will help to identify air convection and advection within the canopy structure. The wind measurements should be carried pairing sonic and cup anemometers at the same locations, allowing to overcome the limitations of sonic

60

anemometers during rain events. The evaporation contribution to the local hydrological cycle by visible vapor plumes requires a detailed quantification of the latent heat flux ($\rho\lambda E$) above and below the canopy. The use of net radiometers at different heights (same locations as wind speed measurements) will complement the detailed air temperature, and wind speed measurements. It is important to underline that some experimental sites worldwide accomplish the equipment requirements above mentioned

65

(FLUXNET, 2020), opening the opportunity to reanalyze their data sets towards the identification of the conditions needed for the formation of visible vapor plumes. Finally, these measurements will benefit from combining high-resolution infrared images from above and below the canopy. These images will provide information during day and night conditions, helping to identify the splash-droplet evaporation process at canopy and ground level when the view field is focused towards specific locations of the forest

70

canopy."

Further clarifications are needed with respect to evaporation estimates - even if used as a proxy in this study (since no direct measurements of the components of the energy balance method were at hand).
**Reply:**

75

We add the following paragraph replacing the section describing the estimation of evaporation carried out by Jimenez-Rodriguez et al (2020):

"An estimation of the evaporation during the monitored period was retrieved from m Jiménez-Rodríguez et al. (2020). This data set is used only as a reference of the evaporation process during the monitoring period on the same site. This because this quantification has limitations

80

accomplishing the Monin-Obukhov similarity (MOST) theory for complex terrains (Breedt et al., 2018). So, it is based only on the vertical transport of water vapor, neglecting the advected energy of the forest canopy."

85

Also, aiming to improve the manuscript quality, we included a series of the changes proposed on the referee's replies. We provide a full list of the main changes done to the manuscript.

Page 1, Line 19 (Paragraph improvement):

"… 2007), where the forest presence at continental scale induced the "biotic pump mechanism"

90

that favored the maintenance of similar precipitation amounts between inland and coastal environments (Makarieva and Gorshkov, 2007; Makarieva et al., 2013a). Meanwhile, the vertical transport …"

Add in Page 4, line 11:

95      "… H21-USB). Meteorological data collected along the tower and soil temperature data were recorded with 1 min and 5 min averages, respectively. All data was summarized in 5 min time intervals for the analysis. A Bushnell® … "

Page 7, Line 3 (Sentence and equation improvements):

100      "… The parameters $\Psi_s$ and $\Psi_L$ were determined with equations 7 and 8, respectively. These equations requires to know the mass of the liquid water in the air ($m_{liq.air}$), the mass of the dry air ($m_{dry.air}$), the density of the air ($\rho_{air}$) and the density of the liquid water content of the air ($\rho_{LWC}$). These variables were determined using the saturation and actual vapor pressures of the air (Stull, 2017). …"

105

$$\Psi_S = \frac{m_{liq.water}}{m_{dry.air}} \qquad (7)$$

$$\Psi_L = \frac{\rho_{LWC}}{\rho_{air}} \qquad (8)$$

110   Add in Page 7, line 15:

"… gradient $\left(\frac{\Delta\theta}{\Delta z}\right)$ due the absence of wind profile measurements to determine the atmospheric stability parameter along the tower. Values of …"

Page 8, Line 29 (New sentence):

115

"…7:00 a.m. Mist might be formed early in the morning during the sampling dates 2018-03-21 and 2018-03-22. However, the time lapse video did not work at those times (Table B1). These mist …"

120   Page 11, Line 18 (Paragraph improvement):

"… As plumes are not stagnant and continue moving upwards thanks to air convection, the water vapor is removed from the understory towards higher altitudes. The water condensation at the canopy level drastically reduced the volume of water vapor due to the phase change 125   (Makarieva et al., 2013b). This allowed the ambient air to remain unsaturated and keeping the "splash droplet evaporation" process providing continuously more water vapor. "

Page 11, line 12:
"… LSBS. Additionally, it cannot be discarded the presence of bioparticles (e.g., airborne 130   bacteria, fungi, pollen, plant fragments, organic compounds) as a source of aerosols from the forests (Huffman et al., 2013, Pöschlet al., 2010, Valsan et al., 2015). The high intensity rains may induce the bioparticles burst from the forest canopy. These bioparticles have been in Australia (Bigg et al., 2015), India (Valsan et al., 2015), Mexico (Rodriguez-Gomez et al., 2020), and the Amazon (Pöschlet al., 2010). Also, convective rains transport from the free troposphere 135   into the boundary layer a portion of the required aerosols …"

Page 7, Line 4:

140 "Due to the lack of instrumentation to estimate the liquid water content in the air, we used a fixed value of 0.05 g m$^{-3}$. This value corresponds to the liquid water in the air reported by Thompson (2007) for continental fog events. The selection of this value was based on (1) the similarity in terms of color and transparency between the vapor plumes and fog, and (2) because both types of events occur close to the ground surface."

145 Page 8, line 25:

"These days were characterized by cumulus clouds crossing the sky above the forest canopy in the day time. Any water vapor ascending from the forest canopy will need to reach a height of more than 100 m to form visible vapor plumes (Figure 3)"

Page 12, line 10:

150 "The description of the formation process of visible vapor plumes provides a first step on the understanding of this phenomenon within forest hydrology. This description helps to identify the timing when this phenomenon occurs, allowing toscreen existing data sets in other tropical research sites to analyze its frequency of occurrence. However, is important to test if the conditions required to form visible vapor plumes are the same in other latitudes and ecosystems. Also, new developments20in air
155 temperature monitoring techniques such as distributed temperature sensing (Euser et al., 2014; Heusinkveld et al., 2020;Izett et al., 2019; Schilperoort et al., 2018) or thermal infrared imagery (Costa et al., 2019; Egea et al., 2017) (Costa et al.,2019, Egea et al., 2017, Nieto et al., 2020, Lapidot et al., 2019) may contribute to accurately quantify the contribution of visible vapor plumes as local recyclers of forest evaporation. These methods are suitable alternatives to eddy-covariance systems that are sensitive to
160 rainy conditions when visible vapor plumes occur."

Also, there is a series of small changes related to grammar or addition of individual words to improve the readability of the manuscript. These changes are not listed in this replied but those are high lined in the new pdf.

[revised manuscript text omitted]

---

## Author Response (AR2)

**Reply to Editor Comments**

In blue we copied the comments of the editor, in black our replies.

Dear authors,

after carefully considering the two (very contrasted) reviews received on your revised manuscript, I would like to invite you to (i) consider the suggestions for technical corrections suggested by referee #1, and (ii) account for the concerns raised by referee #2 (i.e., that the proposed method cannot be applied based on the meteorological measurement setup). The latter point may be addressed by developing (e.g., as part of the discussion/conclusion/outlook) on the need - with a view on potential future follow-up research - for accounting for the limitations of the experimental set-up used in this study and outlining potential avenues for improvements.

I am looking forward to receive an updated version (after minor revisions) of your contribution.

**Reply:**

Dear Editor, aiming to improve the manuscript readability we fix the technical corrections proposed by referee #1 as follows:

1) Page 3, line 4

making difficult to -> making it difficult to

**Reply:** Corrected.

2) Page 4. Line 19, "from 5:00 to 18:30"

In Table B1 the earliest initial time was 5:10 and the latest final time 18:00. Please consider to unify the descriptions.

**Reply:** For this correction, we did the following changes:

We added in page 4, line 11:

> "… 18:30 hours local time (UTC-6). However, the light conditions affected the images selected as suitable for analysis (see Appendix B)."

We updated the caption and footnote of Table B1 as follows:

> **Caption:** "Time windows with suitable images for analysis during the 5 sampling days surveyed with the camera."

> **Note:** "… of the survey. The camera was set to take images from 5:00 to 18:30, the time windows showed in the table correspond to the period with images suitable for analysis."

3) Page 7, line 4

vapor water -> water vapor

**Reply:** Corrected.

4) Page 7, line 15

due the absence -> due to the absence

**Reply:** Corrected.

5) Page 7, line 22

the moist adiabatic -> the dew point temperature
Please delete " at Tz and Tdew.z" because Γdew is independent of the temperatures.

50    **Reply:**  Corrected.

6) Page 8, lines 4 to 5
This because -> This is because
**Reply:**  Corrected.

55

7) Page 12, line 12
However, is -> However, it is
**Reply:**  Corrected.

60    8) Page 12, line 15
Egea et al., 2017) ( -> Egea et al., 2017;
**Reply:**  Corrected.

9) Page 13, line 4
65    de forest -> the forest
**Reply:**  Corrected.

10) Page 13, line 13
fot he -> for the.
70    **Reply:**  Corrected.

Aiming to clarify the concerns expressed by referee #2, we added the following:

1.: A description of the main limitations that current techniques have when measuring fluxes under highly
75    wet atmospheric conditions (Page 12, line 1):

“This paper described the formation of visible vapor plumes based on photographs as a visual
indication of a process that is usually invisible to the human eye. The occurrence of this
phenomenon under rainy conditions makes difficult to quantify its contribution to the forest
80    evaporation with current measuring techniques. Vapor plumes occurrence during rainy days
compromise the performance of more sophisticated instruments that are highly sensitive to rain
or mist conditions Centre for Atmospheric Science, 2020, Mauder and Zeeman, 2018}.
Instruments such as sonic anemometers (e.g., CSAT3, CSAT3B) and Open Path $CO_2$/$H_2O$ Analyzers
(e.g., LI-7500) are strongly affected by high humidity and rainfall Campbell Scientific Inc., 2017,
85    2019, Foken et al., 2012b, LI-COR, 2016, Moncrieff et al., 2005). The presence of rain causes
departures from the measurements increasing the sonic speed (Camuffo 2019, Kelton and
Bricout, 1964, Peters et al., 1998) or blocking the face of the transducers (Campbell Scientific Inc.,
2017) causing a frequency loss during rain events (Zhang et al., 2016). The eddy-covariance
technique is considered as the standard measurement for determining atmospheric fluxes,
90    however, it is dependent on fully turbulent transport over a homogeneous surface (Foken et al.,
2012a). This means that the localized nature of the visible vapor plumes makes measuring them
very susceptible to sensor placement, complicating its monitoring using eddy--covariance systems
located high above the canopy. Additionally, measuring devices based on 3D wind components
(e.g., eddy--covariance systems) are developed to measure water in gas form (Foken et al., 2012a)

95      and are not intended to measure visible vapor plumes that are ascending clusters of tiny water
        particles (Spellman.2012)."

2.: A sentence linking the techniques limitations to further quantification attempts (Page 13, line 11):

100     "… While the quantification of its contribution to the hydrological cycle have to overcome the
        limitations of current measuring techniques."

3.: Following the recommendation of adding potential future follow-up research topics to the manuscript,
we improve two paragraphs with the following additions:
105
Page 13, line 25:
        "… vapor plumes. Also, these sites provide an opportunity to quantify the bias that eddy--
        covariance systems make due to the existence of this phenomenon. Direct measurements of
        atmospheric water (gas and liquid phase) can be achieved with closed-path gas analyzers (e.g, LI-
110     7000DS-LI-COR, EC155-Campbell Sci., FMA-Los Gatos Research), allowing to determine the total
        water content in the air. These measurements will …"

Page 13, line 31:
        "Finally, further research can search for the detailed source of vapor with the implementation of
115     direct measurements of water stable isotopes using mass spectrometers or cavity output
        spectroscopy. This type of research can provide more insights into the effect of vapor plumes on
        the micro-climate of forest ecosystems. Moreover, the occurrence of this phenomenon in other
        vegetation types may be addressed to understand the main drivers and the role played in local
        hydrological systems."
120
We also add a final remark on the conclusion summarising the new additions on the manuscript (see Page
14, line 14):
        "The exploratory nature of this work, opened new research opportunities aiming to improve the
        setup to monitor this phenomenon and provide a further accurate quantification of the
125     contribution within the local hydrology."